# Diffusion & Adversarial Schrödinger Bridges via Iterative Proportional Markovian Fitting

## Abstract

The Iterative Markovian Fitting (IMF) procedure based on iterative reciprocal and Markovian projections has recently been proposed as a powerful method for solving the Schrödinger Bridge problem. However, it has been observed that for the practical implementation of this procedure, it is crucial to alternate between fitting a forward and backward time diffusion at each iteration. Such implementation is thought to be a practical heuristic, which is required to stabilize training and obtain good results in applications such as unpaired domain translation. In our work, we show that this heuristic closely connects with the pioneer approaches for the Schrödinger Bridge based on the Iterative Proportional Fitting (IPF) procedure. Namely, we find that the practical implementation of IMF is, in fact, a combination of IMF and IPF procedures, and we call this combination the Iterative Proportional Markovian Fitting (IPMF) procedure. We show both theoretically and practically that this combined IPMF procedure can converge under more general settings, thus, showing that the IPMF procedure opens a door towards developing a unified framework for solving Schrödinger Bridge problems.

## 1 Introduction

Diffusion models inspired by the Schrödinger Bridge (SB) theory, which connects stochastic processes with the optimal transport theory, have recently emerged as a powerful approach for numerous applications in biology (Tong et al., 2024; Bunne et al., 2023), chemistry (Somnath et al., 2023; Igashov et al.), computer vision (Liu et al., 2023a; Shi et al., 2023) and speech processing (Chen et al., 2023). Most of these applications are dedicated either to supervised translation, e.g., image super-resolution and inpainting (Liu et al., 2023a) or unpaired domain translation, such as image style-transfer (Shi et al., 2023) or single-cell data analysis (Tong et al., 2024).

In this paper, we focus specifically on *unpaired* domain translation, where SB-based algorithms are typically used since they enforce **two** key properties: the similarity between input and translated object (referred to as the *optimality property*) and that the input domain is translated to the target domain (referred to as the *marginal matching property*).

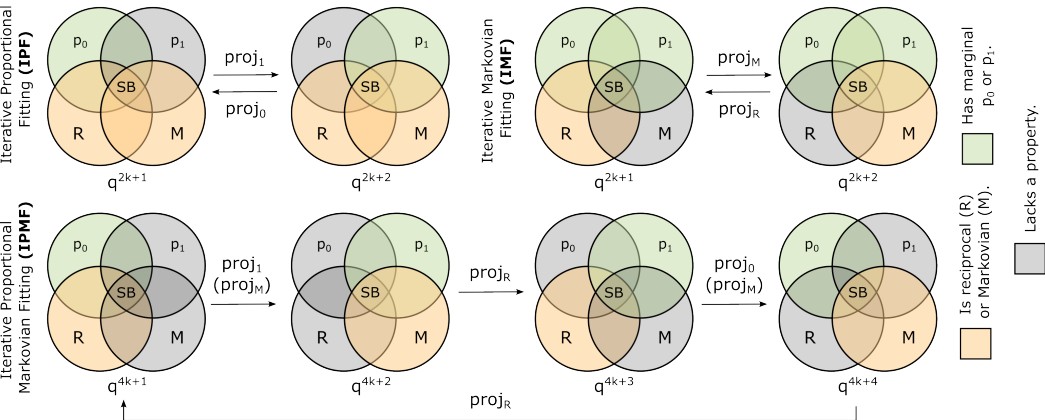

Figure 1: Diagrams of IPF, IMF, and discovered IPMF procedure. All procedures aim to converge to the Schrödinger Bridge, i.e., the reciprocal and Markovian process with marginals $p_0$ and $p_1$.

Early works (De Bortoli et al., 2021; Vargas et al., 2021; Chen et al., 2021) on using Schrödinger Bridge for unpaired domain translation employed the well-celebrated **Iterative Proportional Fitting** (IPF) procedure (Kullback, 1968), also known as the Sinkhorn algorithm (Cuturi & Doucet, 2014). The IPF procedure is initialized with a simple prior process that satisfies the optimality property. It then refines this process iteratively through optimality-preserving transformations until the marginal matching property is achieved. In each iteration, IPF decreases the *forward* KL-divergence $\text{KL}(q^*||q)$ between the current approximation $q$ and the ground-truth Schrödinger Bridge $q^*$. However, due to approximation errors, in practice, IPF may suffer from the "prior forgetting", where the marginal matching property is achieved, but the optimality is lost (Vargas et al., 2024; 2021).

Recently, the **Iterative Markovian Fitting** (IMF) procedure (Shi et al., 2023; Peluchetti, 2023a; Gushchin et al., 2024) was introduced as the promising competitor to IPF. Contrary to IPF, the IMF starts from the stochastic process with the marginal matching property, while the optimality is achieved during the IMF iterations. Unlike the IPF procedure, the IMF procedure at each iteration decreases *reverse* KL-divergence $\text{KL}(q||q^*)$ between the current approximation $q$ and the ground-truth Schrödinger Bridge $q^*$. This iterative procedure can also be seen as a generalization of rectified flows (Liu et al., 2022) to stochastic processes, which is used (Liu et al., 2023b; Yan et al., 2024) for the modern foundational generative models such as Stable Diffusion 3 (Esser et al., 2024). In analogy to IPF, the IMF procedure may also accumulate errors but in approximating data distributions due to a imperfect fit at each iteration, leading to losing marginal matching property.

In practice, the IMF is implemented as a bidirectional procedure alternating between learning forward and backward processes either by diffusion-based models in the **Diffusion** Schrödinger Bridge Matching (DSBM) algorithm (Shi et al., 2023) or GANs in **Adversarial** Schrödinger Bridge Matching (ASBM) algorithm (Gushchin et al., 2024). This heuristic of alternating between learning forward and backward processes helps to stabilize the IMF training and overcome the accumulation of errors and loss of marginal matching property. In this work, we explore the theoretical aspects of this heuristic approach and make the following key contributions.

**Main contributions**: We show that the heuristic bidirectional IMF procedure used in practice closely relates to IPF. Namely, we discover that it, in fact, *secretly* utilizes IPF iterations. Due to this, we propose to call this bidirectional IMF procedure **Iterative Proportional Markovian Fitting** (IPMF) and *conjecture* that it not only converges under more general settings than was previously thought but opens a promising way of developing a unified framework for solving the SB problem.

1. **Theory I.** We prove that the IPMF procedure converges for 1-dimensional Gaussian distributions (§3.2). Even this proof is non-trivial and involves significant complexity, as IPF and IMF minimize different (forward and reverse KL) divergences, leading to interference. We also make and motivate a conjecture about IPMF convergence for multivariate Gaussians (§3.2).

2. **Practice I.** We experimentally support our conjecture for multivariate Gaussians (§4.1), using the closed form update formulas that can be derived in discrete time (§4.1).

3. **Practice II.** We empirically validate through a series of standard experiments, including toy 2D setups (§4.2), the Schrödinger Bridge benchmark (§4.3), colored MNIST images and real-world image dataset - Celeba (§4.4), that IPMF procedure converges.

These contributions show that the IPMF procedure has significant potential to **combine** many previously introduced SB methods, including IPF and IMF-based, with both discrete (Gushchin et al., 2024; De Bortoli et al., 2021) and continuous time (Shi et al., 2023; Peluchetti, 2023a; Vargas et al., 2021), together with their online versions (De Bortoli et al., 2024; Peluchetti, 2024; Karimi et al., 2024). Moreover, the forward-backward framework of IPMF could enable rectified flows to avoid error accumulation, making them even more powerful in diffusion-based generative modeling.

**Notations.** We fix $N \geq 1$ intermediate time moments $0 = t_0 < t_1 < \cdots < t_N < t_{N+1} = 1$ together with $t_0 = 0$ and $t_{N+1} = 1$. We consider discrete stochastic processes with a finite second moment, entropy, and those time-moments as the elements of the set $\mathcal{P}_{2,ac}(\mathbb{R}^{D \times (N+2)})$ of probability distributions on $\mathbb{R}^{D \times (N+2)}$. For any such $q \in \mathcal{P}_{2,ac}(\mathbb{R}^{D \times (N+2)})$, we write $q(x_0, x_{t_1}, \ldots, x_{t_{N+1}})$ to denote its density at a point $(x_0, x_{t_1}, \ldots, x_{t_N}, x_1) \in \mathbb{R}^{D \times (N+2)}$. For convenience we also use the notation $x_{\text{in}} = (x_{t_1}, \ldots, x_{t_N})$ to denote the vector of all intermediate-time variables.

For considering the continuous version of Schrödinger Bridge we denote by $\mathcal{P}(C([0,1]), \mathbb{R}^D)$ the set of continuous stochastic processes with time $t \in [0,1]$, i.e., the set of distributions on continuous trajectories $f : [0,1] \to \mathbb{R}^D$. We use $dW_t$ to denote the differential of the standard Wiener process.

We denote by $p^T \in \mathcal{P}(\mathbb{R}^{D \times (N+2)})$ the discrete process which is the finite-dimensional projection of $T$ to time moments $0 = t_0 < t_1 < \cdots < t_N < t_{N+1} = 1$. In what follows, KL is a short notation for the Kullback-Leibler divergence, while $H$ is a short notation for the differential entropy.

## 2 BACKGROUND

This section recalls the Schrödinger Bridge (SB) problem (§2.1). Then, we describe procedures for solving SB: Iterative Proportional Fitting (IPF, §2.2) and Iterative Markovian Fitting (IMF, §2.3). We discuss the practical implementation of the IMF procedure called Bidirectional IMF in §2.4.

The Schrödinger Bridge problem can be considered both in discrete and continuous time, which are equivalent. The discrete-time and continuous-time versions of IPF and IMF procedures are also similar. In the main text, we operate only with the discrete-time setup to avoid a lot of repetitions, which will harm the flow of the paper. Also, the discrete-time version provides the explicit formulas for the projections used in IPF and IMF procedures, which makes it sufficiently easier to explain the main idea of our paper. We present the analogical facts about the continuous setup in Appendix A.

### 2.1 SCHRÖDINGER BRIDGE (SB) PROBLEM

**Schrödinger Bridge problem formulation.** To formulate the Schrödinger Bridge problem we consider the Wiener process $W^\epsilon$ with the volatility $\epsilon > 0$ which starts at some distribution $p_0$, i.e., the process given by the stochastic differential equation (SDE): $dx_t = \sqrt{\epsilon}dW_t$, $x_0 \sim p_0$. We denote by $W^\epsilon_{|x_0,x_1}$ the stochastic process $W^\epsilon$ conditioned on values $x_0, x_1$ at times $t = 0, 1$, respectively. This process $W^\epsilon_{|x_0,x}$ is called the Brownian Bridge (Ibe, 2013, Chapter 9). The Schrödinger Bridge problem (Schrödinger, 1931) with the Wiener prior between distributions $p_0$ and $p_1$ in the discrete-time setting (De Bortoli et al., 2021) formulates as follows:

$$\min_{q \in \Pi(p_0,p_1)} \text{KL}(q(x_0, x_{\text{in}}, x_1) || p^{W^\epsilon}(x_0, x_{\text{in}}, x_1)), \tag{1}$$

Here $p^{W^\epsilon}(x_0, x_{\text{in}}, x_1)$ is the time-discretization of $W^\epsilon$, which is given by $p^{W^\epsilon}(x_0, x_{\text{in}}, x_1) = p_0(x_0) \prod_{n=1}^{N+1} \mathcal{N}(x_{t_n} | x_{t_{n-1}}, \epsilon(t_n - t_{n-1})I_D)$. In turn, $\Pi(p_0, p_1) \subset \mathcal{P}_{2,ac}(\mathbb{R}^{D \times (N+2)})$ is the subset of discrete-stochastic processes with marginals $q(x_0) = p_0(x_0)$ and $q(x_1) = p_1(x_1)$.

**Static Schrödinger Bridge problem.** One may decompose the objective (1) as follows:

$$\text{KL}(q(x_0, x_{\text{in}}, x_1) || p^{W^\epsilon}(x_0, x_{\text{in}}, x_1)) =$$

$$\text{KL}(q(x_0, x_1) || p^{W^\epsilon}(x_0, x_1)) + \int \text{KL}(q(x_{\text{in}} | x_0, x_1) || p^{W^\epsilon}(x_{\text{in}} | x_0, x_1))q(x_0, x_1)dx_0 dx_1. \tag{2}$$

i.e., KL divergence between $q$ and $p^{W^\epsilon}$ is a sum of two terms: the 1st represents the similarity of the processes' joint, marginal distributions at start and finish times $t = 0, 1$, while the 2nd term represents the average similarity of conditional distributions $q(x_{\text{in}} | x_0, x_1)$ and $p^{W^\epsilon}(x_{\text{in}} | x_0, x_1)$. Since conditional distributions $q(x_{\text{in}} | x_0, x_1)$ can be chosen independently of $q(x_0, x_1)$ we can consider $q(x_{\text{in}} | x_0, x_1) = p^{W^\epsilon}(x_{\text{in}} | x_0, x_1)$. In this case $\text{KL}(q(x_{\text{in}} | x_0, x_1) || p^{W^\epsilon}(x_{\text{in}} | x_0, x_1)) = 0$ for every $x_0, x_1$ and it leads to the Static Schrödinger Bridge problem:

$$\min_{q \in \Pi(p_0,p_1)} \text{KL}(q(x_0, x_1) || p^{W^\epsilon}(x_0, x_1)), \tag{3}$$

In turn, the static SB objective can be expanded as (Gushchin et al., 2023a, Eq. 7):

$$\text{KL}(q(x_0, x_1) || p^{W^\epsilon}(x_0, x_1)) = \int \frac{||x_1 - x_0||^2}{2\epsilon} dq(x_0, x_1) - H(q(x_0, x_1)) + C, \tag{4}$$

which is up to an additive constant is equivalent to the objective of *entropic optimal transport* (EOT) problem with the *quadratic cost* (Cuturi, 2013; Peyré et al., 2019; Léonard, 2013; Genevay, 2019).

### 2.2 ITERATIVE PROPORTIONAL FITTING (IPF)

Several first works on Schrödinger Bridge (Vargas et al., 2021; De Bortoli et al., 2021) propose methods based on the IPF procedure (Kullback, 1968). The IPF-based algorithm is started by setting the process $q^0(x_0, x_{\text{in}}, x_1) = p_0(x_0)p^{W^\epsilon}(x_{\text{in}}, x_1 | x_0)$. Then, the algorithm alternates between two types of IPF projections $\text{proj}_1$ and $\text{proj}_0$ which are given by (De Bortoli et al., 2021, Proposition 2):

$$q^{2k+1} = \text{proj}_1\Big(\underbrace{q^{2k}(x_1) \prod_{n=0}^{N} q^{2k}(x_{t_n} | x_{t_{n+1}})}_{q^{2k}(x_1)q^{2k}(x_0,x_{\text{in}} | x_1)}\Big) \overset{\text{def}}{=} p_1(x_1) \underbrace{\prod_{n=0}^{N} q^{2k}(x_{t_n} | x_{t_{n+1}})}_{q^{2k}(x_0,x_{\text{in}} | x_1)}, \tag{5}$$

$$q^{2k+2} = \text{proj}_0\big(\underbrace{q^{2k+1}(x_0)\prod_{n=1}^{N+1}q^{2k+1}(x_{t_n}|x_{t_{n-1}})}_{q^{2k+1}(x_0)q^{2k+1}(x_{\text{in}},x_1|x_0)}\big) \stackrel{\text{def}}{=} p_0(x_0)\underbrace{\prod_{n=1}^{N+1}q^{2k+1}(x_{t_n}|x_{t_{n-1}})}_{q^{2k+1}(x_{\text{in}},x_1|x_0)}. \quad (6)$$

Thus, these projections replace marginal distributions $q(x_1)$ and $q(x_0)$ in $q(x_0,x_{\text{in}},x_1)$ by $p_1(x_1)$ and $p_0(x_0)$ respectively. This sequence $q^k$ monotonically decreases the forward KL-divergence $\text{KL}(q^*||q^k)$ at each iteration and converges to the solution of the Schrödinger Bridge $q^*$. However, since the prior process $p^{W^\epsilon}$ is used only at the initialization, the imperfect fit in practice at some iteration may lead to deviation from the SB solution. This problem is called "prior forgetting" and was discussed in (Vargas et al., 2024, Appendix E.3). The continuous analog of the IPF procedure is considered in (Vargas et al., 2021) and uses inversions of diffusion processes (see Appendix A.2).

## 2.3 ITERATIVE MARKOVIAN FITTING (IMF)

The Iterative Markovian Fitting (IMF) procedure (Peluchetti, 2023a; Shi et al., 2023) was recently proposed as a strong competitor to the IPF, which does not suffer from the "prior forgetting" problem of IPF. In turn, the discrete-time analog of IMF (D-IMF) has been recently proposed by (Gushchin et al., 2024) to accelerate the inference of the process learned by IMF. The procedure is initialized with any process $q^0 \in \Pi(p_0, p_1)$. Then the procedure alternates between reciprocal $\text{proj}_{\mathcal{R}}$ and Markovian $\text{proj}_{\mathcal{M}}$ projections:

$$q^{2k+1} = \text{proj}_{\mathcal{R}}(q^{2k}) \stackrel{\text{def}}{=} q^{2k}(x_0,x_1)p^{W^\epsilon}(x_{\text{in}}|x_0,x_1), \quad (7)$$

$$q^{2k+2} = \text{proj}_{\mathcal{M}}(q^{2k+1}) \stackrel{\text{def}}{=} \underbrace{q^{2k+1}(x_0)\prod_{n=1}^{N+1}q^{2k+1}(x_{t_n}|x_{t_{n-1}})}_{\text{forward representation}} = \underbrace{q^{2k+1}(x_1)\prod_{n=0}^{N}q^{2k+1}(x_{t_n}|x_{t_{n+1}})}_{\text{backward representation}} \quad (8)$$

Thus, the reciprocal projection $\text{proj}_{\mathcal{R}}(q)$ creates a new (in general, non-Markovian) process by using the joint distribution $q(x_0,x_1)$ and $p^{W^\epsilon}(x_{\text{in}}|x_0,x_1)$. The latter is called the discrete Brownian Bridge. In turn, the Markovian projection $\text{proj}_{\mathcal{M}}(q)$ uses the set of transitional densities $\{q(x_{t_n}|x_{t_{n-1}})\}$ or $\{q(x_{t_n}|x_{t_{n+1}})\}$ to create a new Markovian process starting from $q(x_0)$ or $q(x_1)$ respectively. Unlike the IPF procedure, this sequence $q^k$ monotonically decreases the reverse KL-divergence objective $\text{KL}(q^k||q^*)$ at each iteration and converges to the solution of the Schrödinger Bridge $q^*$. The continuous time version of the IMF procedure is considered in (Shi et al., 2023; Peluchetti, 2023a) and uses similar Markovian and reciprocal projections (Appendix A).

## 2.4 BIDIRECTIONAL IMF

Since the result of the Markovian projection (8) can be represented both by forward and backward representation, in practice, neural networks $\{q_\theta(x_{t_n}|x_{t_{n-1}})\}$ (**forward parametrization**) or $\{q_\phi(x_{t_n}|x_{t_{n+1}})\}$ (**backward parametrization**) are used to learn the corresponding transitional densities. In turn, starting distributions are set to be $q_\theta(x_0) = p_0(x_0)$ for forward parametrization and $q_\phi(x_1) = p_1(x_1)$ for the backward parametrization. In practice, the alternation between forward and backward representations of Markovian processes is used in both implementations of continuous-time IMF by **DSBM** algorithm (Shi et al., 2023, Algorithm 1) based on diffusion models and discrete-time IMF by **ASBM** algorithm (Gushchin et al., 2024, Algorithm 1) based on the GANs. So, this **bidirectional** procedure can be described as follows:

$$q^{4k+1} = \underbrace{q^{4k}(x_0,x_1)p^{W^\epsilon}(x_{\text{in}}|x_0,x_1)}_{\text{proj}_{\mathcal{R}}(q^{4k})}, \quad q^{4k+2} = p(x_1)\underbrace{\prod_{n=0}^{N}q_\phi^{4k+1}(x_{t_{n-1}}|x_{t_n})}_{\text{backward parametrization}}, \quad (9)$$

$$q^{4k+3} = \underbrace{q^{4k+2}(x_0,x_1)p^{W^\epsilon}(x_{\text{in}}|x_0,x_1)}_{\text{proj}_{\mathcal{R}}(q^{4k+1})}, \quad q^{4k+4} = p(x_0)\underbrace{\prod_{n=1}^{N+1}q_\theta^{4k+3}(x_{t_n}|x_{t_{n-1}})}_{\text{forward parametrization}}. \quad (10)$$

Thus, only one marginal is perfectly fitted, e.g., $q_\theta(x_0) = p_0(x_0)$ in the case of forward representation, while the other marginal is only learned, e.g., $q_\theta(x_1) =$

$\int \underbrace{q_\theta(x_0)}_{=p_0(x_0)} \prod_{n=1}^{N+1} q_\theta(x_{t_n}|x_{t_{n-1}}) dx_0 dx_1 \cdots dx_N \approx p_1(x_1)$. It was observed that such approximation errors do not accumulate in this bidirectional version of IMF (Shi et al., 2023; Peluchetti, 2023a; Gushchin et al., 2024), while the usage of only forward or backward parametrization accumulates errors and lead to divergence (De Bortoli et al., 2024, Appendix I).

## 3 ITERATIVE PROPORTIONAL MARKOVIAN FITTING (IPMF)

In this section, we show that the heuristical procedure of bidirectional IMF §2.4 is, in fact, the alternating implementation of both IPF and IMF projections and state that this heuristic in fact defines the new unified Iterative Proportional Markovian Fitting (IPMF) procedure §3.1. Next, in section §3.2, we provide the theoretical analysis of convergence of this IPMF procedure together with the determination of the convergence rate in the case of 1-dimensional Gaussian.

### 3.1 BIDIRECTIONAL IMF IS IPMF

Here, we analyze theoretically what the heuristical bidirectional IMF does. We recall that the IPF projections of $\text{proj}_0(q)$ given by (6) and $\text{proj}_1(q)$ given by (5) of the Markovian process $q$ is in fact just change the starting distribution from $q(x_0)$ to $p_0(x_0)$ and $q(x_1)$ to $p_1(x_1)$. Now we note that the process $q^{4k+2}$ in (9) is obtained by using a combination of Markovian projection $\text{proj}_\mathcal{M}$ given by (8) in forward parametrization and IPF projection $\text{proj}_1$ given by (5):

$$q^{4k+2} = p(x_1) \prod_{n=0}^{N} q^{4k+1}(x_{t_n}|x_{t_{n+1}}) = \text{proj}_1 \big( \underbrace{q^{4k+1}(x_1) \prod_{n=0}^{N} q^{4k+1}(x_n|x_{n+1})}_{\text{proj}_1(\text{proj}_\mathcal{M}(q^{4k+1}))} \big).$$

In turn, the process $q^{4k+4}$ in (10) is obtained by using a combination of Markovian projection $\text{proj}_\mathcal{M}$ given by (8) in backward parametrization and IPF projection $\text{proj}_0$ given by (6):

$$q^{4k+3} = p(x_0) \prod_{n=1}^{N+1} q^{4k+3}(x_{t_n}|x_{t_{n-1}}) = \text{proj}_0 \big( \underbrace{q^{4k+3}(x_0) \prod_{n=1}^{N+1} q^{4k+3}(x_n|x_{n-1})}_{\text{proj}_0(\text{proj}_\mathcal{M}(q^{4k+3}))} \big).$$

Thus, we can represent the bidirectional procedure IMF given by (10) and (9) as follows:

**Iterative Proportional Markovian Fitting (Discrete time setting)**

$$q^{4k+1} = \underbrace{q^{4k}(x_0, x_1) p^{W^\epsilon}(x_{\text{in}}|x_0, x_1)}_{\text{proj}_\mathcal{R}(q^{4k})}, \quad q^{4k+2} = p(x_1) \underbrace{\prod_{n=0}^{N} q^{4k+1}(x_{t_{n-1}}|x_{t_n})}_{\text{proj}_1(\text{proj}_\mathcal{M}(q^{4k+1}))},$$

$$q^{4k+3} = \underbrace{q^{4k+2}(x_0, x_1) p^{W^\epsilon}(x_{\text{in}}|x_0, x_1)}_{\text{proj}_\mathcal{R}(q^{4k+1})}, \quad q^{4k+4} = p(x_0) \underbrace{\prod_{n=1}^{N+1} q^{4k+3}(x_{t_n}|x_{t_{n-1}})}_{\text{proj}_0(\text{proj}_\mathcal{M}(q^{4k+3}))}.$$

Hence, the Bidirectional IMF procedure, in fact, alternates between the two projections of IMF ($\text{proj}_{\mathcal{MR}}$) during which the process "became more optimal" (step towards optimality property) and two IPF projections ($\text{proj}_0$ and $\text{proj}_1$) during which the marginal fitting improves (step towards marginal matching property). Because of it, we have called this (bidirectional IMF) procedure, which starts from any starting process $q^0(x_0, x_{\text{in}}, x_1)$ as **Iterative Proportional Markovian Fitting (IPMF)**. We say that *one IPMF step* consists of these two projections of IMF ($\text{proj}_{\mathcal{MR}}$) and two projections of IPF. We hypothesize that this combined procedure should converge from any starting process $q^0(x_0, x_{\text{in}}, x_1)$, unlike IPF and IMF procedures, which require a specific form of the starting process. In the same time we want to highlight that IPMF becomes IMF if the initial coupling is in the reciprocal class and has the correct marginals $p_0$ and $p_1$. In turn, when the initial coupling is in the Markovian class, reciprocal, and has the correct initial marginal $p_0$ or $p_1$, IPMF becomes IPF. Figure 1 visually illustrates these cases, clarifying the role of the initial coupling and the iterative steps. The analogical analysis of continuous time IPMF is in Appendix A.3.

## 3.2 THEORETICAL ANALYSIS FOR GAUSSIANS

In this section, we analyze the case when $p_0 = \mathcal{N}(\mu_0, \Sigma_0)$ and $p_1 = \mathcal{N}(\mu_1, \Sigma_1)$ are $D$-dimensional Gaussians and the initial process $q^0(x_0, x_{in}, x_1)$ has Gaussian $q^0(x_0, x_1)$ at times $t = 0, 1$. We prove that $q^{4k}(x_0, x_1)$ converges to the solution $q^*(x_0, x_1)$ of the static SB problem (3) for $D = 1$.

We begin with some preparations. We introduce a function $\Xi : \mathbb{R}^{D \times D} \times \mathbb{R}^{D \times D} \times \mathbb{R}^{D \times D} \to \mathbb{R}^{D \times D}$:

$$\Xi(P, \Sigma, \Sigma') \stackrel{\text{def}}{=} (\Sigma')^{-1} P^\top (\Sigma - P(\Sigma')^{-1} P^\top)^{-1}. \tag{11}$$

which is well-defined for $\Sigma \succ 0, \Sigma' \succ 0$ and for $P$ s.t. $\Sigma - P(\Sigma')^{-1} P^\top \succ 0$.

**Lemma 3.1** (Gaussian plans as entropic optimal transport plans). *Consider a $2D$-dimensional Gaussian distribution $q(x_0, x_1) \in \mathcal{P}_{2,ac}(\mathbb{R}^D \times \mathbb{R}^D)$ with marginals $p = \mathcal{N}(\mu, \Sigma)$ and $p' = \mathcal{N}(\mu', \Sigma')$ and correlation $P$ between its components:*

$$q(x_0, x_1) = \mathcal{N}\left( \begin{pmatrix} \mu \\ \mu' \end{pmatrix}, \begin{pmatrix} \Sigma & P \\ P^\top & \Sigma' \end{pmatrix} \right).$$

*Let $A = \Xi(P, \Sigma, \Sigma') \in \mathbb{R}^{D \times D}$. Then $q$ is the unique minimizer of the following problem:*

$$\min_{q' \in \Pi(p, p')} \left\{ \int (-x_1^\top A x_0) \cdot q'(x_0, x_1) dx_0 dx_1 - H(q') \right\}, \tag{12}$$

*where $H(q') = - \int q'(x_0, x_1) \log q'(x_0, x_1) dx_0 dx_1$ is the differential entropy of a distribution.*

Problem (12) is the optimal transport problem with the transport cost $-x_1^\top A x_0$ and entropy regularization (with weight 1), see (Cuturi, 2013; Genevay, 2019). Thus, our lemma states that any Gaussian distribution is a so-called entropic OT plan between its marginals for certain transport cost. In fact, for every Gaussian distribution, we can assign a matrix $A = \Xi(P, \Sigma, \Sigma')$ explaining for which cost this distribution solves the entropic transport problem. We call this matrix the **optimality matrix**. We emphasize that if the optimality matrix is $A = \epsilon^{-1} I_D$, then the transport cost $-\varepsilon^{-1} \cdot \langle x_1, x_0 \rangle$ is equivalent to $\varepsilon^{-1}/2 \cdot \|x_1 - x_0\|^2 = \varepsilon^{-1}/2 \cdot \|x_0\|^2 - \varepsilon^{-1} \cdot \langle x_1, x_0 \rangle + \varepsilon^{-1}/2 \cdot \|x_1\|^2$, and $q$ is the static SB (3) between its marginals for the prior $W^\epsilon$, recall (4).

Now, we make our convergence **conjecture**, which we **theoretically prove for 1-dimensional Gaussians** (Appendix B.4) and experimentally justify for higher dimensions (§4.1).

**Conjecture 3.2** (Quantitative convergence of IPMF for Gaussians). *Let $p_0 = \mathcal{N}(\mu_0, \Sigma_0)$ and $p_1 = \mathcal{N}(\mu_1, \Sigma_1)$ be $D$-dimensional Gaussians. Assume that we run IPMF procedure in the continuous time **or** in discrete time, starting from some $2D$ Gaussian distribution[1]*

$$q^0(x_0, x_1) = \mathcal{N}\left( \begin{pmatrix} \mu_0 \\ \nu \end{pmatrix}, \begin{pmatrix} \Sigma_0 & P_0 \\ P_0 & S_0 \end{pmatrix} \right) \in \mathcal{P}_{2,ac}(\mathbb{R}^D \times \mathbb{R}^D),$$

*and denote the joint distribution obtained after $k$ IPMF steps by*

$$q^{4k}(x_0, x_1) = \mathcal{N}\left( \begin{pmatrix} \mu_0 \\ \nu_k \end{pmatrix}, \begin{pmatrix} \Sigma_0 & P_k \\ P_k & S_k \end{pmatrix} \right).$$

*Denote $A_k \stackrel{\text{def}}{=} \Xi(P_k, \Sigma_0, S_k)$. Then the following bounds hold true:*

$$\|S_k^{-\frac{1}{2}} \Sigma_1 S_k^{-\frac{1}{2}} - I_D\|_2 \leq \alpha^{2k} \|S_0^{-\frac{1}{2}} \Sigma_1 S_0^{-\frac{1}{2}} - I_D\|_2, \quad \|\Sigma_1^{-\frac{1}{2}}(\nu_k - \mu_1)\|_2 \leq \alpha^k \|\Sigma_1^{-\frac{1}{2}}(\nu_0 - \mu_1)\|_2,$$

$$\|A_k - \epsilon^{-1} I_D\|_2 \leq \beta^{2k} \|A_0 - \epsilon^{-1} I_D\|_2, \tag{13}$$

*where $\alpha, \beta < 1$, and $\|\cdot\|_2$ denotes the spectral norm for matrices. The factors $\alpha, \beta$ depend on IPMF type (discrete or continuous), initial parameters $S_0, \nu_0, P_0$, marginal distributions $p_0, p_1$ and $\epsilon$.*

**Justification details.** We find that IPF step keeps the optimality matrix $A_k$ (Lemma B.2) while exponentially improving the marginal matching property of $q(x_0, x_1)$ (Lemma B.1). Next, we analyze closed formulas for IMF step in Gaussian case from (Peluchetti, 2023a; Gushchin et al., 2024). In case $D = 1$, we show that IMF step makes $A_k$ closer to $\frac{1}{\epsilon}$ while not affecting the marginals of $q^{4k}$ (for continuous IMF and discrete IMF with $N = 1$). For higher dimensions, we verify exponential convergence (13) in experiments (§4.1). As a result, IPMF at each round improves both properties.

---

[1]We assume that $q^0(x_0) = p_0(x_0)$, i.e., the initial process starts at $p_0$ at time $t = 0$. This is reasonable, as after the first IPMF round the process will satisfy this property thanks to the IPF projections involved.

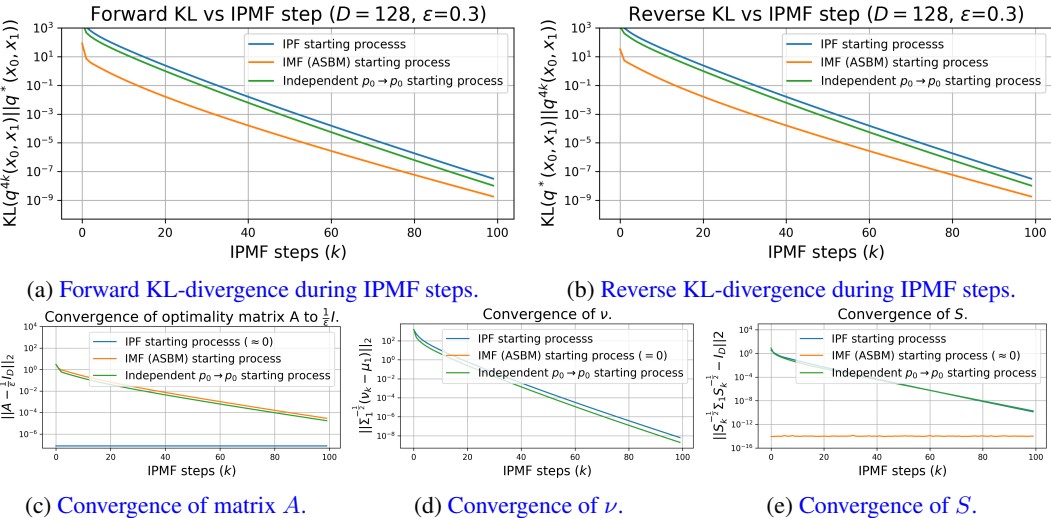

(a) Forward KL-divergence during IPMF steps.    (b) Reverse KL-divergence during IPMF steps.

(c) Convergence of matrix $A$.    (d) Convergence of $\nu$.    (e) Convergence of $S$.

Figure 2: Convergence of IPMF procedure with different starting process $q^0$.

## 4 EXPERIMENTAL ILLUSTRATIONS

In this section, we empirically support that the IPMF procedure converges under a more general setting, specifically for any starting process, unlike IPF and IMF. Thus, our goal is to achieve the same or similar results for all used starting coupling and for both discrete-time (ASBM) and continuous-time (DSBM) solvers. We show in §3.1 and Appendix A that the bidirectional IMF procedure and the introduced IPMF procedure differ only in the initial starting process. Since both practical implementations of continuous-time IMF (Shi et al., 2023, Algorithm 1) and discrete-time IMF (Gushchin et al., 2024, Algorithm 1) use the considered bidirectional version, we use practical algorithms introduced in these works, i.e., Diffusion Schrödinger Bridge Matching (**DSBM**) and Adversarial Schrödinger Bridge Matching (**ASBM**) respectively.

**Experimental setups.** We consider multivariate Gaussian distributions for which we have closed-form IPMF update formulas §4.1, an illustrative 2D example, the Schrödinger Bridges Benchmark (Gushchin et al., 2023b) and real life image data distributions, i.e., the colored MNIST dataset and the Celeba dataset (Liu et al., 2015). All technical details can be found in the Appendix D.

**Starting processes**. In our experiments, we focus on running the IPMF procedure from different initializations, which we call starting processes. We construct starting processes by considering different couplings $q^0(x_0, x_1)$ and using the Brownian Bridge process $W^\epsilon_{|x_0,x_1}$. Thus, for each considered coupling $q^0(x_0, x_1)$ we construct starting process as $q^0(x_0, x_{\text{in}}, x_1) = q^0(x_0, x_1)p^{W^\epsilon}(x_{\text{in}}|x_0, x_1)$ for the discrete-time case and $T^0 = \int W^\epsilon_{|x_0,x_1} dq^0(x_0, x_1)$ for the continuous-time case (Appendix A). For all the experimental setups, we consider the starting processes induced by coupling $q^0(x_0, x_1) = p_0(x_0)p_1(x_1)$, which represent the IMF starting process (used in IMF procedure) and by $q^0(x_0, x_1) = p_0(x_0)p^{W^\epsilon}(x_1|x_0)$, which represent IPF starting process (used in the IPF procedure). We also consider a set of different couplings, which cannot be used either for starting processes of IMF or IPF procedure specifically for each setup to showcase that the IPMF procedure converges under more general assumptions. The results of DSBM and ASBM algorithms starting from different starting processes are denoted as (D/A)SBM-*name of coupling*, e.g., the results for the DSBM using the IMF starting process would be denoted as DSBM-IMF.

### 4.1 HIGH DIMENSIONAL GAUSSIANS

In this section we experimentally validate the convergence of IPMF in the case of the multivariate Gaussian distributions stated in Conjecture 3.2. We conduct experiments using analytical formulas for the Gaussian case for the discrete IMF from (Gushchin et al., 2024, Theorem 3.8). We follow setup from (Gushchin et al., 2023a, Section 5.2) and consider Schrödinger Bridge problem with the dimensionality $D = 128$ and $\epsilon = 0.3$ for centered Gaussians $p_0 = \mathcal{N}(0, \Sigma_0)$ and $p_1 = \mathcal{N}(0, \Sigma_1)$. To construct $\Sigma_0$ and $\Sigma_1$, we sample their eigenvectors from the uniform distribution on the unit sphere and sample their eigenvalues from the log uniform distribution on $[-\log 2, \log 2]$.

We run the IPMF procedure for 100 IPMF steps (each IPMF step consists of 2 IPF projections and two Markovian-Reciprocal projections as stated in §3.1). We use $N = 3$ intermediate time points chosen uniformly between $t = 0$ and $t = 1$. We present in Figure 2 the forward KL-

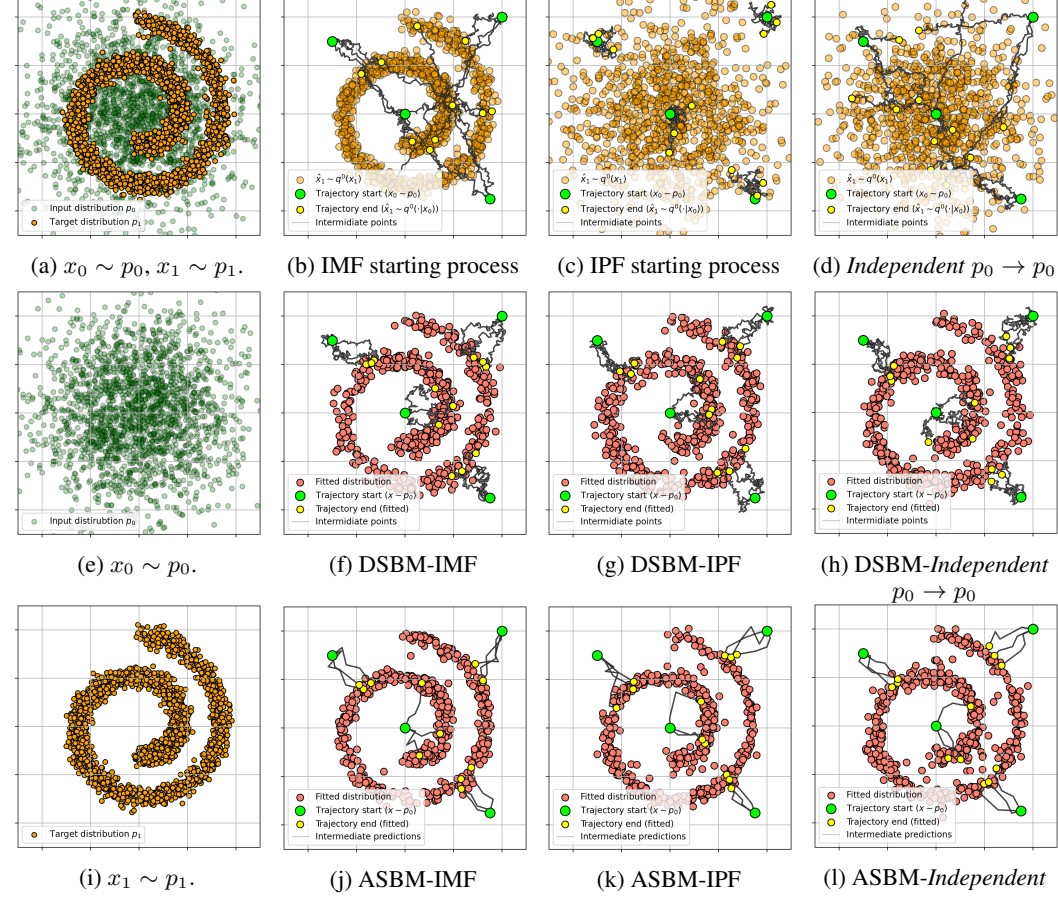

Figure 3: Visualization of learned processes with DSBM and ASBM solvers for *Gaussian→Swiss roll* translation using IMF, IPF, *Independent* $p_0 \to p_0$ starting processes for $\epsilon = 0.1$.

divergence $\text{KL}(q^{4k}(x_0, x_1)||q^*(x_0, x_1))$ and reversed KL-divergence $\text{KL}(q^*(x_0, x_1)||q^{4k}(x_0, x_1))$ between $q^{4k}(x_0, x_1)$ from each IPMF step and the solution of static Schrödinger Bridge $q^*(x_0, x_1)$. For all couplings we observe the exponential convergence in both forward and reverse KL-divergence. We also present the quantities $\|A_k - \epsilon^{-1}I_D\|_2, \|S_k^{-\frac{1}{2}}\Sigma_1 S_k^{-\frac{1}{2}} - I_D\|_2, \|\Sigma_1^{-\frac{1}{2}}(\nu_k - \mu_1)\|_2$ and show they also exhibit the expected behaviour stated in Conjecture 3.2, i.e., converge to zero.

## 4.2 ILLUSTRATIVE 2D EXAMPLE

Here, we consider the SB problem with $\epsilon = 0.1$ with $p_0$ as the 2D Gaussian distribution and $p_1$ as the Swiss roll distribution. As previously mentioned, we train DSBM and ASBM algorithms using IMF and IPF starting processes. Additionally, we consider *Independent* $p_0 \to p_0$ starting processes induced by the coupling $q^0(x_0, x_1) = p_0(x_0)p_0(x_1)$. We present starting processes and results in Figure 3. In all the cases, we observe convergence in the target distribution.

## 4.3 EVALUATION ON THE SB BENCHMARK

We use the SB mixtures benchmark proposed by (Gushchin et al., 2023b) with ground truth solution to the Schrödinger Bridge to test ASBM and DSBM with IMF, IPF and *Independent* $p_0 \to p_0$ (i.e., induced by $q^0(x_0, x_1) = p_0(x_0)p_0(x_1)$ coupling), starting processes.

The benchmark provides continuous distribution pairs $p_0, p_1$ for dimensions $D \in \{2, 16, 64, 128\}$ that have known SB solutions ($q^*$ for discrete setup and $T^*$ for continuous setup), for volatility $\epsilon \in \{0.1, 1, 10\}$. To evaluate the quality of the recovered SB solutions, we use the cB$\mathbb{W}_2^2$-UVP metric as proposed by (Gushchin et al., 2023b) and provide results in Table 1. In addition we study how all the approaches learn the target distribution in Appendix C.

As can be seen, DSBM and ASBM starting from all the processes at $\epsilon \in \{1, 10\}$ yield quite similar results, but on the $\epsilon = 0.1$ DSBM and ASBM with IPF and *Independent* $p_0 \to p_0$ starting processes metric do experience a slight decrease.

| | Algorithm Type | $\epsilon = 0.1$ | | | | $\epsilon = 1$ | | | | $\epsilon = 10$ | | | |
|---|---|---|---|---|---|---|---|---|---|---|---|---|---|
| | | $D=2$ | $D=16$ | $D=64$ | $D=128$ | $D=2$ | $D=16$ | $D=64$ | $D=128$ | $D=2$ | $D=16$ | $D=64$ | $D=128$ |
| Best algorithm on benchmark† | Varies | 1.94 | 13.67 | 11.74 | 11.4 | 1.04 | 9.08 | 18.05 | 15.23 | 1.40 | 1.27 | 2.36 | **1.31** |
| DSBM-IMF | | 1.21 | 4.61 | 9.81 | 19.8 | 0.68 | **0.63** | 5.8 | 29.5 | 0.23 | 5.45 | 68.9 | 362 |
| DSBM-IPF | | 2.55 | 17.4 | 15.85 | 17.45 | 0.29 | 0.76 | 4.05 | 29.59 | 0.35 | 3.98 | 83.2 | 210 |
| DSBM-$Ind(p_0, p_0)$ | | 2.72 | 11.7 | 16.5 | 17.02 | 0.41 | 0.92 | **3.7** | 29 | 0.16 | 3.91 | 101 | 255 |
| ASBM-IMF† | IPMF | 0.89 | 8.2 | 13.5 | 53.7 | **0.19** | 1.6 | 5.8 | **10.5** | **0.13** | 0.4 | 1.9 | 4.7 |
| ASBM-IPF | | 3.06 | 14.37 | 44.35 | 32.5 | 0.18 | 1.68 | 9.25 | 20.47 | **0.13** | **0.36** | 2.28 | 4.97 |
| ASBM-$Ind(p_0, p_0)$ | | 3.99 | 15.73 | 39.3 | 40.32 | 0.18 | 1.68 | 6.16 | 12.8 | **0.13** | 0.38 | **1.36** | **2.6** |
| SF$^2$M-Sink† | | **0.54** | **3.7** | **9.5** | **10.9** | 0.2 | 1.1 | 9 | 23 | 0.31 | 4.9 | 319 | 819 |

Table 1: Comparisons of cB$\mathbb{W}_2^2$-UVP $\downarrow$ (%) between the static SB solution $q^*(x_0, x_1)$ and the learned solution on the SB benchmark. The best metric is **bolded**. Results marked with † are taken from (Gushchin et al., 2024) and (Gushchin et al., 2023b). The results of DSBM and ASBM algorithms starting from different starting processes are denoted as (D/A)SBM-*name of starting process*

## 4.4 UNPAIRED IMAGE-TO-IMAGE TRANSLATION

To test our approach on real data, we consider two unpaired image-to-image translation setups: *colorized 3 → colorized 2* digits from the MNIST dataset with 32×32 resolution size and *male→female* faces from the Celeba dataset with 64×64 resolution size.

**Colored MNIST**. We construct train and test sets by RGB colorization of MNIST digits from corresponding train and test sets of classes "2" and "3". We train ASBM and DSBM algorithms starting from the IMF process. In addition, we test starting process induced by the independent coupling of the distribution of colored digits of class "3" ($p_0$) and the distribution of colored digits of class "7" with inverted RGB channels ($p^{inv7}(x_1)$), we call this process *Inverted 7*, i.e., $q^0(x_0, x_1) = p_0(x_0)p^{inv7}(x_1)$. We visualize the *Inverted 7* starting the process in Figure 4. Further technical details can be found in the Appendix D. We learn DSBM and ASBM on the *train* set of digits and visualize the translated *test* images in Figure 5.

We observe that both DSBM and ASBM algorithms starting from both IMF and *Inverted 7* starting process fit the

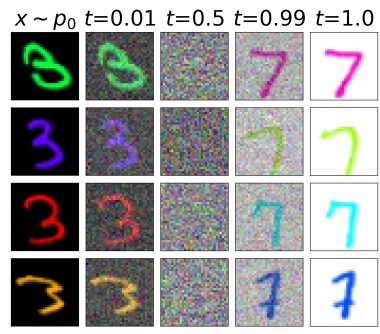

Figure 4: *Inverted 7* starting process, i.e., reciprocal process with marginals $p_0$ and $p^{inv7}$, visualization.

target distribution of colored MNIST digits of class "2" and preserve the color of the input image during translation. This supports that the IPMF procedure converges to the same solution, which resembles the solution of the Schrödinger Bridge.

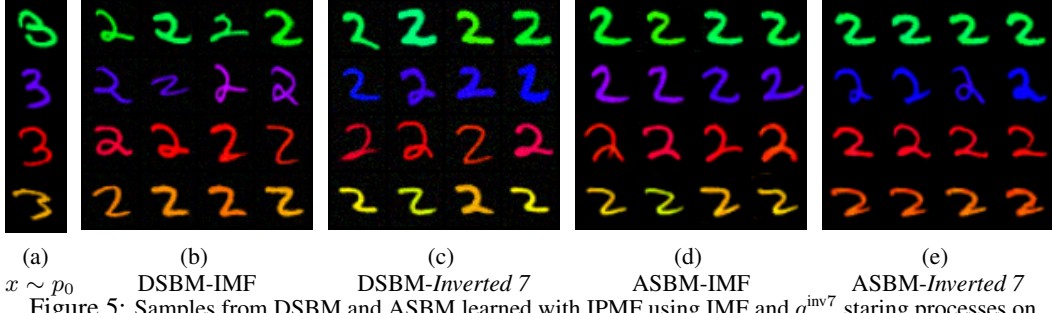

| (a) | (b) | (c) | (d) | (e) |
|---|---|---|---|---|
| $x \sim p_0$ | DSBM-IMF | DSBM-*Inverted 7* | ASBM-IMF | ASBM-*Inverted 7* |

Figure 5: Samples from DSBM and ASBM learned with IPMF using IMF and $q^{inv7}$ staring processes on Colored MNIST *3→2* (32 × 32) translation for $\epsilon = 10$.

**Celeba.** In this setup, we consider the variation of the IMF starting process called IMF-OT, where the starting process is induced by mini batch optimal transport coupling $q^{OT}(x_0, x_1)$ (Tong et al., 2024), and *Independent $p_0 \to p_0$* starting process. In addition, we test the DSBM algorithm with starting processes induced by *DDPM SDEdit* and *SD SDEdit* couplings, which is the SDEdit method Meng et al. (2021) used for *male→female* translation with 1) DDPM Ho et al. (2020) model trained on the female part of Celeba and 2) Stable Diffusion v1.5 Rombach et al. (2022) with designed text prompt, more details are provided in Appendix C.2 including generated examples in Figure 8. We use approximately the same number of parameters for DSBM and ASBM generator and use 10% of male and female images as the test set for evaluation, for other details see Appendix D. We provide qualitative results for IMF-OT and *Independent $p_0 \to p_0$* starting processes in Figure 7 and quantitative analysis through plotting FID as function of number of IPMF iterations in Figure 6.

We see from the qualitative results in Figure 7 that presented models: 1) converge to the target distribution, 2) keep alignment between the features of the input images and generated images (e.g.,

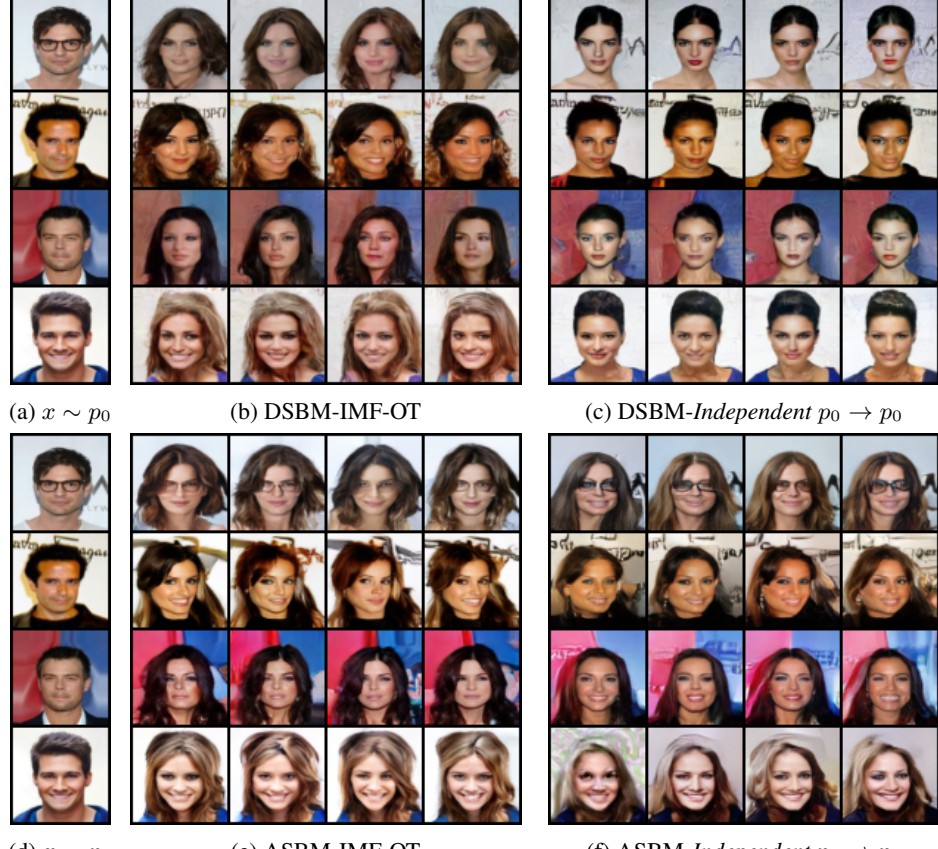

(a) $x \sim p_0$     (b) DSBM-IMF-OT     (c) DSBM-*Independent* $p_0 \rightarrow p_0$

(d) $x \sim p_0$     (e) ASBM-IMF-OT     (f) ASBM-*Independent* $p_0 \rightarrow p_0$

Figure 7: Results of CelebA at 64×64 size for *male→female* translation learned with ASBM and DSBM using IMF-OT and *Independent* $p_0 \rightarrow p_0$ starting processes for $\epsilon = 1$.

the color of hair, background e.t.c.). It should be noted, however, that the samples generated by models differ, because different starting processes lead to different neural network optimization trajectories, and as a result some of the starting processes give a better fit to the target distribution, see FID plot in Figure 6, and some better preserve the input image features, see MSE plot in Figure 9b.. From the FID plot Figure6, we see that despite different starting processes and continuous or discrete time settings of DSBM and ASBM, all the models fit the target distribution quite well.

## 5 DISCUSSION

**Potential impact.** The presented Iterative Proportional Markovian Fitting procedure shows a potential to overcome the error accumulation problem observed in distillation methods like rectified flows (Liu et al., 2022; 2023b), which is used for the acceleration of the foundational image generation models, e.g. StableDiffusion 3 in (Esser et al., 2024). These distillation methods are based on the one-directional IMF procedure in the limit of Schrödinger Bridge hyperparameter $\epsilon = 0$. However, the one-directional version is observed to accumulate errors and may even lead to the divergence (De Bortoli et al., 2024, Appendix I). Furthermore, using the limiting case of $\epsilon = 0$ makes it impossible to use the IPF procedure to restore marginals. The usage of a bidirectional version along with $\epsilon > 0$ should both correct the

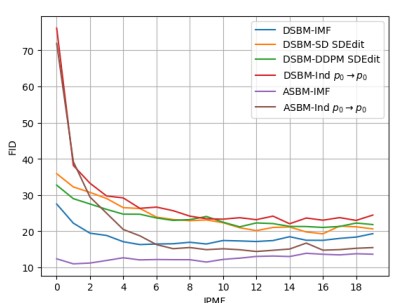

Figure 6: Convergence of models to target distribution. FID plotted as a function of IPMF iterations for all the presented setups.

marginals and make trajectories of diffusion more straight to accelerate the inference of diffusion models. We believe that considering such distillation techniques from the IPMF point of view may help to overcome the current limitations of these techniques.

**Limitations.** While we show the proof of exponential convergence of the IPMF procedure for the 1-dimensional Gaussians in the continuous-time IMF and discrete-time IMF (with one inner point $N = 1$), and present a wide set of experiments supporting this procedure, the proof of convergence of IPMF in the general case still remains a promising avenue for future work.

**Reproducibility Statement**. For all experiments presented, the full set of hyperparameters is shown either in Section 4 or in Appendix D. In addition, the code is submitted as a supplementary material with guidelines how to run every experiment are included. Derivations supporting theoretical claims Lemma 3.1 and Conjecture 3.2 in case $D = 1$ are included in the Appendix B.

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

# A    CONTINUOUS-TIME SCHRÖDINGER BRIDGE SETUP

## A.1    SCHRÖDINGER BRIDGE (SB) PROBLEM IN CONTINUOUS-TIME

This section covers the continuous-time formulation of SB as its IPF and IMF procedures. First, we introduce several new notations to better align the continuous version with the discrete-time version considered in the main text. Consider the Markovian process $T$ defined by the corresponding forward or backward (time-reversed) SDEs:

$$T : dx_t = v^+(x_t, t)dt + \sqrt{\epsilon}dW_t^+, \quad x_0 \sim p_0(x_0),$$
$$T : dx_t = v^-(x_t, t)dt + \sqrt{\epsilon}dW_t^-, \quad x_1 \sim p_1(x_1),$$

where we additionally denote by $W_t^+$ and $W_t^-$ the Wiener process in forward or backward time. We say $T_{|x_0}$ and $T_{|x_1}$ denotes the conditional process of $T$ fixing the marginals using delta functions, i.e. setting $p_0(x_0) = \delta_{x_0}(x)$ and $p_1(x_1) = \delta_{x_1}(x)$:

$$T_{|x_0} : dx_t = v^+(x_t, t)dt + \sqrt{\epsilon}dW_t^+, \quad x_0 \sim \delta_{x_0}(x),$$
$$T_{|x_1} : dx_t = v^-(x_t, t)dt + \sqrt{\epsilon}dW_t^-, \quad x_1 \sim \delta_{x_1}(x).$$

Moreover, we use $p(x_0)T_{|x_0}$ to denote the stochastic process which starts by sampling $x_0 \sim p(x_0)$ and then moving this $x_0$ according the SDE given by $T_{|x_0}$, i.e., $p(x_0)T_{|x_0}$ is short for the process $\int T_{|x_0}p(x_0)dx_0$. Finally, we use the shortened notation of the process $T_{|0,1}(x_0, x_1)$ conditioned

on its values at times 0 and 1, saying $p^T(x_0, x_1)T_{|0,1}(x_0, x_1) = \int T_{|0,1}(x_0, x_1)p^T(x_0, x_1)dx_0dx_1$. This visually links the following equations with the discrete-time formulation.

**Schrödinger Bridge problem.** Considering the continuous case, the Schrödinger Bridge problem is stated using continuous stochastic processes instead of one with predefined timesteps. Thus, the Schrödinger Bridge problem finds the most likely in the sense of Kullback-Leibler divergence stochastic process $T$ with respect to prior Wiener process $W^\epsilon$, i.e.:

$$\min_{T \in \mathcal{F}(p_0, p_1)} \mathrm{KL}(T||W^\epsilon), \tag{14}$$

where $\mathcal{F}(p_0, p_1) \subset \mathcal{P}(C([0,1]), \mathbb{R}^D)$ is the set of all stochastic processes pinned marginal distribution $p_0$ and $p_1$ at times 0 and 1, respectively. The minimization problem (14) has a unique solution $T^*$ which can be represented as forward or backward diffusion (Léonard, 2013):

$$T^* : dx_t = v^{*+}(x_t, t)dt + \sqrt{\epsilon}dW_t^+, \quad x_0 \sim p_0(x_0),$$
$$T^* : dx_t = v^{*-}(x_t, t)dt + \sqrt{\epsilon}dW_t^-, \quad x_1 \sim p_1(x_1),$$

where $v^{*+}$ and $v^{*-}$ are the corresponding drift functions.

**Static Schrödinger Bridge problem.** As in discrete-time, Kullback-Leibler divergence in (14) could be decomposed as follows:

$$\mathrm{KL}(T||W^\epsilon) = \mathrm{KL}(p^T(x_0, x_1)||p^{W^\epsilon}(x_0, x_1)) + \int \mathrm{KL}(T_{|x_0, x_1}||W_{|x_0, x_1}^\epsilon)dp^T(x_0, x_1). \tag{15}$$

It has been proved (Léonard, 2013) that for the solution $T^*$ it's conditional process is given by $T_{|x_0, x_1}^* = W_{|x_0, x_1}^\epsilon$. Thus, we can set $T_{|x_0, x_1} = W_{|x_0, x_1}^\epsilon$ zeroing the second term in (15) and minimize over processes with $T_{|x_0, x_1} = W_{|x_0, x_1}^\epsilon$. This leads to the equivalent Static formulation of the Schrödinger Bridge problem:

$$\min_{q \in \Pi(p_0, p_1)} \mathrm{KL}(q(x_0, x_1)||p^{W^\epsilon}(x_0, x_1)), \tag{16}$$

where $\Pi(p_0, p_1)$ is the set of all joint distributions with marginals $p_0$ and $p_1$. Whether time is discrete or continuous, the decomposition of SB leads to the same static formulation, which, is closely related to Entropic OT as shown in (4).

## A.2 ITERATIVE PROPORTIONAL FITTING (IPF) FOR CONTINUOUS-TIME

Following the main text, we describe the IPF procedure for continuous-time setup using stochastic processes. Likewise, IPF starts with setting $T^0 = p_0(x_0)W_{|x_0}^\epsilon$ and then it alternates between following projections:

$$T^{2k+1} = \mathrm{proj}_1\left(p^{T^{2k}}(x_1)T_{|x_1}^{2k}\right) \overset{\text{def}}{=} p_1(x_1)T_{|x_1}^{2k}, \tag{17}$$

$$T^{2k+2} = \mathrm{proj}_0\left(p^{T^{2k+1}}(x_0)T_{|x_0}^{2k+1}\right) \overset{\text{def}}{=} p_0(x_0)T_{|x_0}^{2k+1}. \tag{18}$$

As in the discrete-time case, these projections replace marginal distributions $p^T(x_1)$ and $p^T(x_0)$ in the processes $p^T(x_1)T_{|x_1}$ and $p^T(x_0)T_{|x_0}$ by $p_1(x_1)$ and $p_0(x_0)$ respectively. Similarly to discrete-time formulation, the sequence of $T^k$ converges to the solution of the Schrödinger Bridge problem $T^*$ implicitly the reverse Kullback-Leibler divergence $\mathrm{KL}(T^k||T^*)$ between the current process $T^k$ and the solution to the SB problem $T*$. Additionally, it should be mentioned that projections are conducted by numerical approximation of forward and time-reversed conditional processes, $T_{|x_0}$ and $T_{|x_1}$, by learning their drifts via one of the methods: score matching (De Bortoli et al., 2021) or maximum likelihood estimation (Vargas et al., 2021).

## A.3 ITERATIVE MARKOVIAN FITTING (IMF) FOR CONTINUOUS-TIME

IMF introduces new projections that alternate between reciprocal and Markovian processes starting from any process $T^0$ pinned by $p_0$ and $p_1$, i.e. in $\mathcal{F}(p_0, p_1)$:

$$T^{2k+1} = \mathrm{proj}_\mathcal{R}\left(T^{2k}\right) \overset{\text{def}}{=} p^{T^{2k}}(x_0, x_1)W_{|x_0, x_1}^\epsilon, \tag{19}$$

$$T^{2k+2} = \text{proj}_{\mathcal{M}}\left(T^{2k+1}\right) \stackrel{\text{def}}{=} \underbrace{p^{T^{2k+1}}(x_0)T^{2k+1}_{M|x_0}}_{\text{forward representation}} = \underbrace{p^{T^{2k+1}}(x_1)T^{2k+1}_{M|x_1}}_{\text{backward representation}}. \tag{20}$$

where we denote by $T_M$ the Markovian projections of the processes $T$, which can be represented as the forward or backward time diffusion as follows (Gushchin et al., 2024, Section 2.1):

$$T_M : dx_t^+ = v_M^+(x_t^+, t)dt + \sqrt{\epsilon}dW_t^+, \quad x_0 \sim p^T(x_0), \quad v_M^+(x_t^+, t) = \int \frac{x_1 - x_t^+}{1 - t}p^T(x_1|x_t)dx_1,$$

$$T_M : dx_t^- = v_M^-(x_t^-, t)dt + \sqrt{\epsilon}dW_t^-, \quad x_1 \sim p^T(x_1), \quad v_M^-(x_t^-, t) = \int \frac{x_0 - x_t^-}{1 - t}p^T(x_0|x_t)dx_0.$$

This procedure converges to a unique solution, which is the Schrödinger bridge $T^*$ (Léonard, 2013). While reciprocal projection can be easily done by combining the joint distribution $p^T(x_0, x_1)$ of the process $T$ and Brownian bridge $W^\epsilon_{|x_0,x_1}$, the Markovian projection is much more challenging and must be fitted via Bridge matching (Shi et al., 2023; Liu et al.; Peluchetti, 2023b).

Since the result of the Markovian projection can be represented (8) both by forward and backward representation, in practice, neural networks $v_\theta^+$ (**forward parametrization**) or $v_\phi^-$ (**backward parametrization**) are used to learn the corresponding drifts of the Markovian projections. In turn, starting distributions are set to be $p_0(x_0)$ for forward parametrization and $p_1(x_1)$ for the backward parametrization. So, this **bidirectional** procedure can be described as follows:

$$T^{4k+1} = \underbrace{p^{T^{4k}}(x_0, x_1)W^\epsilon_{|x_0,x_1}}_{\text{proj}_{\mathcal{R}}(T^{4k})}, \quad T^{4k+2} = \underbrace{p_1(x_1)T^{4k+1}_{M|x_1}}_{\text{backward parametrization}}, \tag{21}$$

$$T^{4k+3} = \underbrace{p^{T^{4k+2}}(x_0, x_1)W^\epsilon_{|x_0,x_1}}_{\text{proj}_{\mathcal{R}}(T^{4k+2})}, \quad T^{4k+4} = \underbrace{p_0(x_0)T^{4k+3}_{M|x_0}}_{\text{forward parametrization}}. \tag{22}$$

### A.4 Iterative Proportional Markovian Fitting (IPMF) for continuous-time

Here, we analyze the continuous version of the heuristical bidirectional IMF. First, we recall, that the IPF projections $\text{proj}_0(T)$ and $\text{proj}_1(T)$ given by (17) and (18) of the Markovian process $T$ is just change the starting distribution from $p^T(x_0)$ to $p_0(x_0)$ and $p^T(x_1)$ to $p_1(x_1)$.

Now we note that the process $T^{4k+2}$ in (21) is obtained by using a combination of Markovian projection $\text{proj}_{\mathcal{M}}$ given by (20) in forward parametrization and IPF projection $\text{proj}_1$ given by (18):

$$T^{4k+2} = p_1(x_1)T^{4k+1}_{M|x_1} = \underbrace{\text{proj}_1\left(p^{T^{4k+1}}(x_1)T^{4k+1}_{M|x_1}\right)}_{\text{proj}_1(\text{proj}_{\mathcal{M}}(T^{4k+1}))}.$$

In turn, the process $T^{4k+4}$ in (22) is obtained by using a combination of Markovian projection $\text{proj}_{\mathcal{M}}$ given by (20) in backward parametrization and IPF projection $\text{proj}_0$ given by (17):

$$T^{4k+3} = p_0(x_0)T^{4k+3}_{M|x_0} = \underbrace{\text{proj}_0\left(p^{T^{4k+3}}(x_0)T^{4k+3}_{M|x_0}\right)}_{\text{proj}_0(\text{proj}_{\mathcal{M}}(T^{4k+3}))}.$$

Combining these facts we can rewrite bidirectional IMF in the following manner:

**Iterative Proportional Markovian Fitting (Conitnious time setting)**

$$T^{4k+1} = \underbrace{p^{T^{4k}}(x_0, x_1)W^\epsilon_{|x_0,x_1}}_{\text{proj}_{\mathcal{R}}(T^{4k})}, \quad T^{4k+2} = \underbrace{p_1(x_1)T^{4k+1}_{M|x_1}}_{\text{proj}_1(\text{proj}_{\mathcal{M}}(T^{4k+1}))} \tag{23}$$

$$T^{4k+3} = \underbrace{p^{T^{4k+2}}(x_0, x_1)W^\epsilon_{|x_0,x_1}}_{\text{proj}_{\mathcal{R}}(T^{4k+2})}, \quad T^{4k+4} = \underbrace{p_0(x_0)T^{4k+3}_{M|x_0}}_{\text{proj}_0(\text{proj}_{\mathcal{M}}(T^{4k+3}))}. \tag{24}$$

Thus, we obtain the analog of the discrete-time IPMF procedure, which concludes our description of the continuous setups.

## B    THEORETICAL ANALYSIS FOR GAUSSIANS

Here we study behavior of IPMF with parameter $\varepsilon$ between $D$-dimensional Gaussians $p_0 = \mathcal{N}(\mu_0, \Sigma_0)$ and $p_1 = \mathcal{N}(\mu_1, \Sigma_1)$. For convenience, we use notation $\varepsilon_*$ instead of $\varepsilon$. For $D = 1$, we prove that the parameters of $q^{4k}$ with each step exponentially converge to desired values $\mu_0, \mu_1, \Sigma_0, \Sigma_1, \varepsilon_*$ for continuous and discrete (with $N = 1$) IMF. The steps are as follows:

**1)** In Appendix B.1, we reveal the connection between $2D$-dimensional Gaussian distribution and solution of entropic OT problem with specific transport cost, i.e., we prove our Lemma 3.1.

**2)** In Appendix B.2, we study the effect of IPF steps on the current process. We show that during these steps, the marginals become close to $p_0$ and $p_1$, while the optimality matrix does not change.

**3)** In Appendix B.3, we study the effect of IMF step on the current process when $D = 1$. We show that after IMF (continuous or discrete with $N = 1$), marginals remain the same, while the new correlation becomes close to the correlation of the static $\varepsilon_*$-EOT solution between marginals.

**4)** Finally, in Appendix B.4, we prove our main Conjecture 3.2 for case of continuous or discrete (with $N = 1$) in dimension $D = 1$.

### B.1    GAUSSIAN PLANS AS ENTROPIC OPTIMAL TRANSPORT PLANS

*Proof of Lemma 3.1.* We note that we can add any functions $f(x_0)$ and $g(x_1)$ depending only on $x_0$ or $x_1$, respectively, to the cost function $c(x_0, x_1) = x_1^\top A x_0$, and the OT solution will not change. This is because the integrals of such functions over any transport plan will be constants as they will depends only on the marginals (which are given) but not on the plan itself. Thus, for any $A \in \mathbb{R}^{D \times D}$, we can rearrange the cost term $c(x_0, x_1)$ so that it becomes lower-bounded:

$$\tilde{c}(x_0, x_1) = \|A x_0\|^2/2 - x_1^\top A x_0 + \|x_1\|^2/2 = \|A x_0 - x_1\|^2/2 \geq 0,$$

where $\tilde{c}(x_0, x_1)$ is a lower bounded function. Following (Gushchin et al., 2023b, Theorem 3.2), the conditional distribution $q_c(x_1|x_0)$ with the lower bounded cost function $c$ can be expressed as:

$$q_c(x_0|x_1) \quad \propto \quad \exp\left(-c(x_0, x_1) + f_c(x_0)\right) = \exp\left(x_1^\top A x_0 + f_c(x_0)\right),$$

where function $f_c(x_1)$ depends only on $x_1$. Moreover, we can simplify this distribution to

$$q_c(x_0|x_1) \quad = \quad Z_{x_0} Z_{x_1} \exp\left(x_1^\top A x_0\right), \tag{25}$$

where factors $Z_{x_0}$ and $Z_{x_1}$ depend only on $x_0$ and $x_1$, respectively. Meanwhile, the conditional distribution of $q(x_0|x_1)$ has a closed form, namely,

$$\begin{aligned} q(x_0|x_1) \quad &= \quad \mathcal{N}\left(x_0|\mu' + P(\Sigma')^{-1}(x_1 - \mu'), \Sigma - P(\Sigma')^{-1}P^\top\right) \\ &= \quad Z_{x_0} Z_{x_1} \exp\left(x_0^\top (\Sigma - P(\Sigma')^{-1}P^\top)^{-1} P(\Sigma')^{-1} x_1\right). \end{aligned}$$

where factors $Z_{x_0}$ and $Z_{x_1}$ depend only on $x_0$ and $x_1$, respectively. Equating terms of $q(x_0|x_1)$ and $q_c(x_0|x_1)$ which depend on $x_0$ and $x_1$ simultaneously, we obtain the required function

$$A = (\Sigma')^{-1} P^\top (\Sigma - P(\Sigma')^{-1} P^\top)^{-1}, \tag{26}$$

which concludes that $q$ solves 1-entropic OT with the cost function $-x_0^\top A x_1$. $\qquad \square$

### B.2    IPF STEP ANALYSIS

We use IPMF with parameter $\varepsilon_*$ between distributions $\mathcal{N}(\mu_0, \Sigma_0)$ and $\mathcal{N}(\mu_1, \Sigma_1)$. We start with the process $\mathcal{N}\left(\begin{pmatrix} \mu_0 \\ \nu \end{pmatrix}, \begin{pmatrix} \Sigma_0 & P \\ P & S \end{pmatrix}\right)$ with correlation $P$.

Recall that one IPMF step consists of 4 consecutive steps:

1. IMF step refining current optimality value,
2. IPF step changing final prior to $\mathcal{N}(\mu_1, \Sigma_1)$,
3. IMF step refining current optimality value,

4. IPF step changing starting prior to $\mathcal{N}(\mu_0, \Sigma_0)$.

We use the following notations for the covariance matrices changes during IPMF step:

$$\begin{pmatrix} \Sigma_0 & P \\ P^T & S \end{pmatrix} \xRightarrow{IMF} \begin{pmatrix} \Sigma_0 & \tilde{P} \\ (\tilde{P})^T & S \end{pmatrix} \xRightarrow{IPF} \begin{pmatrix} (S') & P' \\ (P')^T & \Sigma_1 \end{pmatrix}$$

$$\xRightarrow{IMF} \begin{pmatrix} S' & \hat{P} \\ (\hat{P})^T & \Sigma_1 \end{pmatrix} \xRightarrow{IPF} \begin{pmatrix} \Sigma_0 & P'' \\ (P'')^T & S'' \end{pmatrix},$$

and for the means the changes are:

$$\begin{pmatrix} \mu_0 \\ \nu \end{pmatrix} \xRightarrow{IMF} \begin{pmatrix} \mu_0 \\ \nu \end{pmatrix} \xRightarrow{IPF} \begin{pmatrix} \nu' \\ \mu_1 \end{pmatrix} \xRightarrow{IMF} \begin{pmatrix} \nu' \\ \mu_1 \end{pmatrix} \xRightarrow{IPF} \begin{pmatrix} \mu_0 \\ \nu'' \end{pmatrix}.$$

**Lemma B.1** (Improvement after IPF steps). *Consider an initial 2D-dimensional Gaussian joint distribution* $\mathcal{N}\left( \begin{pmatrix} \mu_0 \\ \nu \end{pmatrix}, \begin{pmatrix} \Sigma_0 & P \\ P^T & S \end{pmatrix} \right) \in \mathcal{P}_{2,ac}(\mathbb{R}^D \times \mathbb{R}^D)$. *We run IPMF step between distributions* $\mathcal{N}(\mu_0, \Sigma_0)$ *and* $\mathcal{N}(\mu_1, \Sigma_1)$ *and obtain new joint distribution* $\mathcal{N}\left( \begin{pmatrix} \mu_0 \\ \mu'' \end{pmatrix}, \begin{pmatrix} \Sigma_0 & P'' \\ (P'')^\top & S'' \end{pmatrix} \right)$. *Then, the distance between ground truth* $\mu_1, \Sigma_1$ *and the new joint distribution parameters decreases as:*

$$\|(S'')^{-\frac{1}{2}} \Sigma_1 (S'')^{-\frac{1}{2}} - I_D\|_2 \quad \leq \quad \|\tilde{P}_n\|_2^2 \cdot \|P_n''\|_2^2 \cdot \|S^{-\frac{1}{2}} \Sigma_1 S^{-\frac{1}{2}} - I_D\|_2, \tag{27}$$

$$\|\Sigma_1^{-\frac{1}{2}} (\nu'' - \mu_1)\|_2 \quad \leq \quad \|\hat{P}_n^\top\|_2 \cdot \|P_n'\|_2 \cdot \|\Sigma_1^{-\frac{1}{2}} (\nu - \mu_1)\|_2, \tag{28}$$

*where* $\tilde{P}_n := \Sigma_0^{-1/2} \tilde{P} S^{-1/2}, P_n' := (S')^{-\frac{1}{2}} P' \Sigma_1^{-\frac{1}{2}}, \hat{P}_n := (S')^{-1/2} \hat{P} \Sigma_1^{-1/2}$ *and* $P_n'' := \Sigma_0^{-1/2} P'' (S'')^{-1/2}$ *are normalized matrices whose spectral norms are not greater than* 1.

*Proof.* During IPF steps, we keep the conditional distribution and change the marginal. For the first IPF, we keep the inner part $x_0|x_1$ for all $x_1 \in \mathbb{R}^D$:

$$\mathcal{N}\left( x_0 | \mu_0 + \tilde{P} S^{-1}(x_1 - \nu), \Sigma_0 - \tilde{P} S^{-1} \tilde{P}^\top \right) = \mathcal{N}\left( x_0 | \nu' + P' \Sigma_1^{-1}(x_1 - \mu_1), S' - P' \Sigma_1^{-1}(P')^\top \right).$$

This is equivalent to the system of equations:

$$\Sigma_0 - \tilde{P} S^{-1} \tilde{P}^\top = S' - P' \Sigma_1^{-1}(P')^\top, \tag{29}$$

$$P' \Sigma_1^{-1} = \tilde{P} S^{-1}, \tag{30}$$

$$\mu_0 - \tilde{P} S^{-1} \nu = \nu' - P' \Sigma_1^{-1} \mu_1. \tag{31}$$

Similarly, after the second IPF step, we have equations:

$$\Sigma_1 - \hat{P}^\top (S')^{-1} \hat{P} = S'' - (P'')^\top \Sigma_0^{-1} P'', \tag{32}$$

$$(P'')^\top \Sigma_0^{-1} = \hat{P}^\top (S')^{-1}, \tag{33}$$

$$\mu_1 - \hat{P}^\top (S')^{-1} \nu' = \nu'' - (P'')^\top \Sigma_0^{-1} \mu_0. \tag{34}$$

**Covariance matrices.** Combining equations (30), (29) and (33), (32) together, we obtain:

$$\Sigma_0 - S' = \tilde{P} S^{-1}(S - \Sigma_1) S^{-1} \tilde{P}^\top, \qquad //(33),(32) \tag{35}$$

$$I_D - \Sigma_0 (S')^{-1} = \tilde{P} S^{-1}(\Sigma_1 - S) S^{-1} \tilde{P}^\top (S')^{-1}, \qquad //(35) \cdot (S')^{-1} \tag{36}$$

$$\Sigma_1 - S'' = \hat{P}^\top (S')^{-1}(I_D - \Sigma_0 (S')^{-1}) \hat{P}, \qquad //(30),(29) \tag{37}$$

$$\Sigma_1 - S'' = \hat{P}^\top (S')^{-1} \tilde{P} S^{-1}(\Sigma_1 - S) S^{-1} \tilde{P}^\top (S')^{-1} \hat{P}, \qquad //(36) \text{ insert to } (37)$$

$$\Sigma_1 - S'' = (P'')^\top \Sigma_0^{-1} \tilde{P} S^{-1}(\Sigma_1 - S) S^{-1} \tilde{P}^\top \Sigma_0^{-1} P'', \qquad //\text{change using } (33)$$

$$(S'')^{-\frac{1}{2}} \Sigma_1 (S'')^{-\frac{1}{2}} - I_D = (S'')^{-\frac{1}{2}} (P'')^\top \Sigma_0^{-\frac{1}{2}} \cdot \Sigma_0^{-\frac{1}{2}} \tilde{P} S^{-\frac{1}{2}} \cdot (S^{-\frac{1}{2}} \Sigma_1 S^{-\frac{1}{2}} - I_D) \cdot S^{-\frac{1}{2}} \tilde{P}^\top \Sigma_0^{-\frac{1}{2}} \cdot \Sigma_0^{-\frac{1}{2}} P'' (S'')^{-\frac{1}{2}}.$$

The matrices (29) and (32) must be SPD to be covariance matrices:

$$\Sigma_0 - \tilde{P} S^{-1} \tilde{P}^\top \succeq 0 \quad \implies \quad I_D \succeq \Sigma_0^{-1/2} \tilde{P} S^{-1/2} \cdot S^{-1/2} \tilde{P}^T \Sigma_0^{-1/2},$$

$$S'' - (P'')^\top \Sigma_0^{-1} P'' \succeq 0 \implies I_D \succeq \Sigma_0^{-1/2} P'' (S'')^{-1/2} \cdot (S'')^{-1/2} (P'')^\top \Sigma_0^{-1/2}.$$

In other words, denoting matrices $\tilde{P}_n := \Sigma_0^{-1/2} \tilde{P} S^{-1/2}$ and $P''_n := \Sigma_0^{-1/2} P'' (S'')^{-1/2}$, we can bound their spectral norms as $\|\tilde{P}_n\|_2 \le 1$ and $\|P''_n\|_2 \le 1$. We write down the final transaction for covariance matrices:

$$(S'')^{-\frac{1}{2}} \Sigma_1 (S'')^{-\frac{1}{2}} - I_D = (P''_n)^\top \cdot \tilde{P}_n \cdot (S^{-\frac{1}{2}} \Sigma_1 S^{-\frac{1}{2}} - I_D) \cdot \tilde{P}_n^\top \cdot P''_n. \tag{38}$$

Hence, the spectral norm of the difference between ground truth $\Sigma_1$ and current $S''$ drops exponentially as:

$$\|(S'')^{-\frac{1}{2}} \Sigma_1 (S'')^{-\frac{1}{2}} - I_D\|_2 \le \|\tilde{P}_n\|_2^2 \cdot \|P''_n\|_2^2 \cdot \|S^{-\frac{1}{2}} \Sigma_1 S^{-\frac{1}{2}} - I_D\|_2.$$

**Means.** Combining equations (31), (30) and (34), (33) together, we obtain:

$$\begin{aligned}
\mu_0 - \nu' &= \tilde{P} S^{-1} \nu - P' \Sigma_1^{-1} \mu_1 = P' \Sigma_1^{-1} (\nu - \mu_1), & //(31),(30) \quad (39) \\
\nu'' - \mu_1 &= (P'')^\top \Sigma_0^{-1} \mu_0 - \hat{P}^\top (S')^{-1} \nu' = \hat{P}^\top (S')^{-1} (\mu_0 - \nu'), & //(34),(33) \quad (40) \\
\nu'' - \mu_1 &= \hat{P}^\top (S')^{-1} P' \Sigma_1^{-1} (\nu - \mu_1), & //\text{insert } (39) \text{ to } (40) \\
\Sigma_1^{-\frac{1}{2}} (\nu'' - \mu_1) &= \Sigma_1^{-\frac{1}{2}} \hat{P}^\top (S')^{-\frac{1}{2}} \cdot (S')^{-\frac{1}{2}} P' \Sigma_1^{-\frac{1}{2}} \cdot \Sigma_1^{-\frac{1}{2}} (\nu - \mu_1). &
\end{aligned}$$

The matrices (29) and (32) must be SPD to be covariance matrices:

$$\begin{aligned}
S' - P' \Sigma_1^{-1} (P')^\top \succeq 0 &\implies I_D \succeq (S')^{-\frac{1}{2}} P' \Sigma_1^{-\frac{1}{2}} \cdot \Sigma_1^{-\frac{1}{2}} (P')^\top (S')^{-\frac{1}{2}}, \\
\Sigma_1 - \hat{P}^\top (S')^{-1} \hat{P} \succeq 0 &\implies I_D \succeq \Sigma_1^{-1/2} \hat{P}^\top (S')^{-1/2} \cdot (S')^{-1/2} \hat{P} \Sigma_1^{-1/2}.
\end{aligned}$$

Denoting matrices $P'_n := (S')^{-\frac{1}{2}} P' \Sigma_1^{-\frac{1}{2}}$ and $\hat{P}_n := (S')^{-1/2} \hat{P} \Sigma_1^{-1/2}$, we can bound their spectral norms as $\|P'_n\|_2 \le 1$ and $\|\hat{P}_n\|_2 \le 1$. We use this to estimate the $\ell_2$-norm of the difference between the ground truth $\mu_1$ and the current mean:

$$\Sigma_1^{-\frac{1}{2}} (\nu'' - \mu_1) = \hat{P}_n^\top \cdot P'_n \cdot \Sigma_1^{-\frac{1}{2}} (\nu - \mu_1), \tag{41}$$

$$\|\Sigma_1^{-\frac{1}{2}} (\nu'' - \mu_1)\|_2 \le \|\hat{P}_n^\top\|_2 \cdot \|P'_n\|_2 \cdot \|\Sigma_1^{-\frac{1}{2}} (\nu - \mu_1)\|_2.$$

$\square$

**Lemma B.2** (IPF step does not change optimality matrix $A$)**.** *Consider an initial 2D-dimensional Gaussian joint distribution* $\mathcal{N}\left(\begin{pmatrix} \mu_0 \\ \nu \end{pmatrix}, \begin{pmatrix} \Sigma_0 & \tilde{P} \\ \tilde{P}^\top & S \end{pmatrix}\right) \in \mathcal{P}_{2,ac}(\mathbb{R}^D \times \mathbb{R}^D)$. *We run IPF step between distributions* $\mathcal{N}(\mu_0, \Sigma_0)$ *and* $\mathcal{N}(\mu_1, \Sigma_1)$ *and obtain new joint distribution* $\mathcal{N}\left(\begin{pmatrix} \nu' \\ \mu_1 \end{pmatrix}, \begin{pmatrix} S' & P' \\ P' & \Sigma_1 \end{pmatrix}\right)$. *Then, IPF step does not change optimality matrix $A$, i.e.,*

$$A = \Xi(P, \Sigma_0, S) = \Xi(P', S', \Sigma_1).$$

*Proof.* The explicit formulas for $\Xi(\tilde{P}, \Sigma_0, S)$ and $\Xi(P', S', \Sigma_1)$ are

$$\begin{aligned}
\Xi(\tilde{P}, \Sigma_0, S) &= S^{-1} \tilde{P}^\top \cdot (\Sigma_0 - \tilde{P} S^{-1} \tilde{P}^\top), \\
\Xi(P', S', \Sigma_1) &= \Sigma_1^{-1} (P')^\top \cdot (S' - P' \Sigma_1^{-1} (P')^\top).
\end{aligned}$$

The first terms are equal due to equation (30), and the second terms are equal due to (29).

We can prove this lemma in more general way. We derive the formula (26) for $A$ only from the shape of the conditional distribution $q(x_0|x_1)$ (25). During IPF step, this distribution remains the same by design, while parameters $S, \tilde{P}$ change. Hence, IPF step has no effect on the optimality matrix. $\square$

### B.3 IMF STEP ANALYSIS IN $1D$

**Preliminaries.** In case $D = 1$, we work with scalars $\mu_0, \mu_1, \sigma_0^2, \sigma_1^2$ instead of matrices $\mu_0, \mu_1, \Sigma_0, \Sigma_1$. The correlation matrix $P$ can be restated as $P = \rho\sigma_0\sigma_1$, where $\rho \in (-1, 1)$ is the correlation coefficient. Using these notations, formula (11) for optimality coefficient $\chi \in \mathbb{R}$ (instead of matrix $A$) can be expressed as

$$\Xi(\rho, \sigma, \sigma') \stackrel{\text{def}}{=} \frac{\rho}{\sigma\sigma'(1 - \rho^2)} = \chi \in (-\infty, +\infty). \tag{42}$$

The function $\Xi$ is monotonously increasing w.r.t. $\rho \in (-1, 1)$ and, thus, invertible, i.e., there exists a function $\Xi^{-1} : (-\infty, +\infty) \times \mathbb{R}_+ \times \mathbb{R}_+ \to (-1, 1)$ such that

$$\Xi^{-1}(\chi, \sigma, \sigma') = \frac{\sqrt{\chi^2\sigma^2(\sigma')^2 + 1/4} - 1/2}{\chi\sigma\sigma'}. \tag{43}$$

The inverse function is calculated via solving quadratic equation w.r.t. $\rho$.

In our paper, we consider both discrete and continuous IMF. By construction, IMF step does change marginals of the process it works with. However, for both continuous and discrete IMF, the new correlation converges to the correlation of the $\varepsilon_*$-EOT between marginals.

**Lemma B.3** (Correlation improvement after (D)IMF step). *Consider a 2-dimensional Gaussian distribution with marginals $p = \mathcal{N}(\mu, \sigma^2)$ and $p' = \mathcal{N}(\mu', (\sigma')^2)$ and correlation $\rho \in (-1, 1)$ between its components. After continuous IMF or DIMF with single time point $t$, we obtain correlation $\rho_{new}$. The distance between $\rho_{new}$ and EOT correlation $\rho_* = \Xi^{-1}(1/\varepsilon_*, \sigma, \sigma')$ decreases as:*

$$|\rho_{new} - \rho_*| \quad \leq \quad \gamma \cdot |\rho - \rho_*|,$$

*where factor $\gamma$ for continuous and discrete IMF (with $N = 1$) is, respectively,*

$$\gamma_c(\sigma, \sigma') = \frac{\sqrt{\sigma^2(\sigma')^2 + \varepsilon_*^2/4} - \varepsilon_*/2}{\sigma\sigma'}, \tag{44}$$

$$\gamma_d(\sigma, \sigma', t) = \frac{1}{1 + \frac{t^2(1-t)^2\sigma^2(\sigma')^2 + t(1-t)(t^2(\sigma')^2 + (1-t)^2\sigma^2)\varepsilon + t^2(1-t)^2\varepsilon_*^2}{(1-t)^2((1-t)^2\sigma^2 + t\sigma\sigma')^2 + t^2(t(\sigma')^2 + (1-t)\sigma\sigma')^2 + t(1-t)((1-t)\sigma + t\sigma')^2\varepsilon_*}}. \tag{45}$$

*Proof.* **Continuous case.** Following (Peluchetti, 2023a, Eq. 42), we have the formula for $\rho_{new}$:

$$\rho_{new}(\rho) = \exp\left\{-\varepsilon_* \frac{\tanh^{-1}\left(\frac{c_1}{c_3}\right) + \tanh^{-1}\left(\frac{c_2}{c_3}\right)}{c_3}\right\} > 0, \tag{46}$$

$$c_1 = \varepsilon_* + 2(\sigma')^2(\rho\sigma^2 - (\sigma')^2), c_3 = \sqrt{(\varepsilon_* + 2(\rho+1)\sigma^2(\sigma')^2)(\varepsilon_* + 2(\rho-1)\sigma^2(\sigma')^2)},$$

$$c_2 = \varepsilon_* + 2\sigma^2(\rho(\sigma')^2 - \sigma^2).$$

The map $\rho_{new}(\rho)$ is contraction over $\rho \in [-1, 1]$ with the contraction coefficient $\gamma_c(\sigma, \sigma') \stackrel{\text{def}}{=} \frac{\sqrt{\sigma^2(\sigma')^2 + \varepsilon_*^2/4} - \varepsilon_*/2}{\sigma\sigma'} < 1$. The unique fixed point of such map is $\rho_* = P(1/\varepsilon_*, \sigma_0, \sigma_1)$, since IMF does not change $\varepsilon_*$-EOT solution. Hence, we derive a bound

$$|\rho_{new}(\rho) - \rho_*| = |\rho_{new}(\rho) - \rho_{new}(\rho_*)| \leq \gamma_c(\sigma, \sigma')|\rho - \rho_*|.$$

**Discrete case** ($N = 1$). In this case, we use notations from (Gushchin et al., 2024), namely, we denote covariance matrix $\begin{pmatrix} \Sigma_0 & P \\ P & \Sigma_1 \end{pmatrix} \stackrel{\text{def}}{=} \begin{pmatrix} \sigma^2 & \rho\sigma\sigma' \\ \rho\sigma\sigma' & (\sigma')^2 \end{pmatrix}$.

The general formulas of DIMF step are given for time points $0 = t_0 < t_1 < \cdots < t_N < t_{N+1} = 1$. Following (Gushchin et al., 2024), we have an explicit formula for reciprocal step. For any $1 \leq k \leq N$, we have joint covariance between time moments

$$\Sigma_{t_k, t_k} = (1 - t_k)^2\Sigma_0 + 2t_k(1 - t_k)P + t_k^2\Sigma_1 + t_k(1 - t_k)\varepsilon_*,$$

$$\Sigma_{t_{k+1}, t_k} = (1 - t_k)(1 - t_{k+1})\Sigma_0 + [(1 - t_k)t_{k+1} + (1 - t_{k+1})t_k]P + t_k t_{k+1}\Sigma_1 + t_k(1 - t_{k+1})\varepsilon_*,$$

$$\Sigma_{t_1, 0} = (1 - t_1)\Sigma_0 + t_1 P,$$

$$\Sigma_{1,t_N} = t_N\Sigma_1 + (1 - t_N)P.$$

Matrices $\Sigma_{t_{k+1},t_k}$ and $\Sigma_{t_k,t_k}$ depend on $P$. For Markovian step, we write down an analytical formula for the new covariance $\Sigma_{new}$ between marginals:

$$f(P) \stackrel{\text{def}}{=} P_{new}(P) = \prod_{k=0}^{N} \left(\Sigma_{t_{k+1},t_k} \cdot \Sigma_{t_k,t_k}^{-1}\right) \cdot \Sigma_0,$$

The derivative of $f'(P)$ is as follows:

$$
\begin{aligned}
f'(P) &= \left[\frac{(1 - t_N)}{\Sigma_{1,t_N}} - \frac{2t_N(1 - t_N)}{\Sigma_{t_N,t_N}} + \frac{t_1}{\Sigma_{t_1,0}}\right] \cdot f(P) \\
&+ \sum_{k=1}^{N-1} \left[\frac{[(1 - t_k)t_{k+1} + t_k(1 - t_{k+1})]}{\Sigma_{t_{k+1},t_k}} - \frac{2t_k(1 - t_k)P}{\Sigma_{t_k,t_k}}\right] \cdot f(P) \\
&= \sum_{k=0}^{N} \left[-\frac{t_{k+1}(1 - t_{k+1})}{\Sigma_{t_{k+1},t_{k+1}}} + \frac{t_{k+1}(1 - t_k) + t_k(1 - t_{k+1})}{\Sigma_{t_{k+1},t_k}} - \frac{t_k(1 - t_k)}{\Sigma_{t_k,t_k}}\right] \cdot f(P) \quad (47)
\end{aligned}
$$

In the case of single point $t = t_1$ ($N = 1$), we prove that the function $f(P)$ is a contraction map. The sufficient condition for the map to be contraction is to have derivative's norm bounded by $\gamma_d < 1$.

Firstly, we can write down the simplified formula $\rho_{new}(\rho)$ in our original notations:

$$\rho_{new}(\rho) = \frac{((1 - t)\sigma + t\rho\sigma')(t\sigma' + (1 - t)\rho\sigma)}{(1 - t)\sigma^2 + 2t(1 - t)\rho\sigma\sigma' + t^2(\sigma')^2 + t(1 - t)\varepsilon_*}. \quad (48)$$

Next, we simplify derivative (47):

$$
\begin{aligned}
\Sigma_{0,t} &= (1 - t) \cdot \Sigma_0 + t \cdot P, \\
\Sigma_{t,1} &= t \cdot \Sigma_1 + (1 - t) \cdot P, \\
\Sigma_{t,t} &= (1 - t)^2 \cdot \Sigma_0 + 2(1 - t)t \cdot P + t^2 \cdot \Sigma_1 + t(1 - t)\varepsilon_0 = (1 - t) \cdot \Sigma_{0,t} + t \cdot \Sigma_{t,1} + t(1 - t)\varepsilon, \\
f'(P) &= \frac{(1 - t)\Sigma_{0,t}}{\Sigma_{t,t}} + \frac{t\Sigma_{t,1}}{\Sigma_{t,t}} - 2 \cdot \frac{t\Sigma_{t,1} \cdot (1 - t)\Sigma_{0,t}}{\Sigma_{t,t} \cdot \Sigma_{t,t}}.
\end{aligned}
$$

We define new variables $\tilde{\Sigma}_{0,t} \stackrel{\text{def}}{=} (1 - t)\Sigma_{0,t}$, $\tilde{\Sigma}_{1,t} \stackrel{\text{def}}{=} t\Sigma_{t,1}$ and $\tilde{\varepsilon}_* = t(1 - t)\varepsilon_*$. We note that while $P \in [-\sqrt{\Sigma_0\Sigma_1}, \sqrt{\Sigma_0\Sigma_1}]$ the value $\tilde{\Sigma}_{0,t} + \tilde{\Sigma}_{1,t} = (1 - t)^2 \cdot \Sigma_0 + 2(1 - t)t \cdot P + t^2 \cdot \Sigma_1 \geq 0$. Then, we restate $f'$ as:

$$
\begin{aligned}
f' &= \frac{\tilde{\Sigma}_{0,t}}{\tilde{\Sigma}_{0,t} + \tilde{\Sigma}_{1,t} + \tilde{\varepsilon}_*} + \frac{\tilde{\Sigma}_{1,t}}{\tilde{\Sigma}_{0,t} + \tilde{\Sigma}_{1,t} + \tilde{\varepsilon}_*} - \frac{2\tilde{\Sigma}_{0,t}\tilde{\Sigma}_{1,t}}{(\tilde{\Sigma}_{0,t} + \tilde{\Sigma}_{1,t} + \tilde{\varepsilon}_*)^2} \quad (49) \\
&= \frac{(\tilde{\Sigma}_{0,t} + \tilde{\Sigma}_{1,t})(\tilde{\Sigma}_{0,t} + \tilde{\Sigma}_{1,t} + \tilde{\varepsilon}_*) - 2\tilde{\Sigma}_{0,t}\tilde{\Sigma}_{1,t}}{(\tilde{\Sigma}_{0,t} + \tilde{\Sigma}_{1,t} + \tilde{\varepsilon}_*)^2} \\
&= \frac{\tilde{\Sigma}_{0,t}^2 + \tilde{\Sigma}_{1,t}^2 + (\tilde{\Sigma}_{0,t} + \tilde{\Sigma}_{1,t})\tilde{\varepsilon}_*}{(\tilde{\Sigma}_{0,t} + \tilde{\Sigma}_{1,t} + \tilde{\varepsilon}_*)^2} \quad (50) \\
&= \frac{\tilde{\Sigma}_{0,t}^2 + \tilde{\Sigma}_{1,t}^2 + (\tilde{\Sigma}_{0,t} + \tilde{\Sigma}_{1,t})\tilde{\varepsilon}_*}{\tilde{\Sigma}_{0,t}^2 + 2\tilde{\Sigma}_{0,t}\tilde{\Sigma}_{1,t} + \tilde{\Sigma}_{1,t}^2 + 2(\tilde{\Sigma}_{0,t} + \tilde{\Sigma}_{1,t})\tilde{\varepsilon}_* + \tilde{\varepsilon}_*^2} \quad (51) \\
&= \frac{1}{1 + \frac{2\tilde{\Sigma}_{0,t}\tilde{\Sigma}_{1,t} + (\tilde{\Sigma}_{0,t} + \tilde{\Sigma}_{1,t})\tilde{\varepsilon}_* + \tilde{\varepsilon}_*^2}{\tilde{\Sigma}_{0,t}^2 + \tilde{\Sigma}_{1,t}^2 + (\tilde{\Sigma}_{0,t} + \tilde{\Sigma}_{1,t})\tilde{\varepsilon}_*}}. \quad (52)
\end{aligned}
$$

We note that all terms in (50) are greater than 0:

$$0 < f'(P), \quad P \in [-\sqrt{\Sigma_0\Sigma_1}, \sqrt{\Sigma_0\Sigma_1}]. \quad (53)$$

In negative segment $P \in [-\sqrt{\Sigma_0\Sigma_1}, 0]$, the derivative $f'$ is greater than in positive segment $[0, \sqrt{\Sigma_0\Sigma_1}]$, and edge value $f(-\sqrt{\Sigma_0\Sigma_1}) > -\sqrt{\Sigma_0\Sigma_1}$. Thus, in negative segment, the convergence to the fixed point $\rho_*\sqrt{\Sigma_0\Sigma_1} > 0$ is faster, than in positive segment.

For $P \in [0, \sqrt{\Sigma_0 \Sigma_1}]$, we can bound the fraction in denominator of (52) by taking its numerator's minimum at $P = 0$ and its denominator's maximum at $P = \sqrt{\Sigma_0 \Sigma_1}$, i.e,

$$0 < f' \leq \gamma_d(\Sigma_0, \Sigma_1, t) < 1,$$

$$\gamma_d(\Sigma_0, \Sigma_1, t) = \frac{1}{1 + \frac{t^2(1-t)^2 \Sigma_0 \Sigma_1 + t(1-t)(t^2 \Sigma_1 + (1-t)^2 \Sigma_0)\varepsilon + t^2(1-t)^2 \varepsilon_*^2}{(1-t)^2((1-t)\Sigma_0 + t\sqrt{\Sigma_0\Sigma_1})^2 + t^2(t\Sigma_1 + (1-t)\sqrt{\Sigma_0\Sigma_1})^2 + t(1-t)((1-t)\sqrt{\Sigma_0} + t\sqrt{\Sigma_1})^2 \varepsilon_*}}.$$

We note that $\gamma_d(\Sigma_0, \Sigma_1, t)$ is increasing function w.r.t. $\Sigma_0, \Sigma_1$.

If we put into the function $f$ argument $P_* = \rho_* \sqrt{\Sigma_0 \Sigma_1}$ corresponding to the $\varepsilon_*$-EOT correlation, DIMF does not change it. Hence, $P_*$ is the fixed point of $f(P)$, and we have

$$|P_{new} - P_*| = |f(P) - f(P_*)| \leq \gamma_d(\Sigma_0, \Sigma_1, t)|\Sigma - \Sigma_*|.$$

Dividing both sides by $\sqrt{\Sigma_0 \Sigma_1}$, we get

$$|\rho_{new} - \rho_*| \leq \gamma_d(\Sigma_0, \Sigma_1, t)|\rho - \rho_*|.$$

$\square$

**Lemma B.4** ($\chi$ improvement after (D)IMF step). *Consider a 2-dimensional Gaussian distribution with marginals $p = \mathcal{N}(\mu, \sigma^2)$ and $p' = \mathcal{N}(\mu', (\sigma')^2)$ and correlation $\rho \in (-1, 1)$ between its components. After continuous IMF or DIMF with a single time point $t$, we obtain new correlation $\rho_{new}$, such that $|\rho_{new} - \rho_*| \leq \gamma|\rho - \rho_*|$ where $\rho_* = P(1/\varepsilon_*, \sigma, \sigma')$ and $\gamma < 1$ is from (44) for IMF and from (45) for DIMF. We have bound in terms of $\chi = \Xi(\rho, \sigma, \sigma')$ and $\chi_{new} = \Xi(\rho_{new}, \sigma, \sigma')$:*

$$|\chi_{new} - 1/\varepsilon_*| \leq l(\rho, \rho_*, \gamma) \cdot |\chi - 1/\varepsilon_*|, \tag{54}$$

$$l(\rho, \rho_*, \gamma) = \left[1 - (1 - \gamma)\frac{(1 - \max\{\rho_*, |\rho|\}^2)^2}{1 + \max\{\rho_*, |\rho|\}^2}\right] < 1.$$

*Proof.* **Monotone.** The function $\rho_{new}$ from (46) for continuous IMF and from (48) for DIMF is monotonously increasing on $(-1, 1)$. For continuous IMF, with the growth of $\rho$, $c_3$ grows faster than $c_1$ or $c_2$, hence, $\frac{c_1}{c_3}, \frac{c_2}{c_3}$ and $\tanh^{-1}\left(\frac{c_1}{c_3}\right), \tanh^{-1}\left(\frac{c_2}{c_3}\right)$ decrease. Thus, the power in the exponent of $\rho_{new}$ and $\rho_{new}$ itself increase. For DIMF, the derivative of $\rho_{new}$ greater than zero based on (53).

The monotone means that the value $\rho_{new}$ always remains from the same side from $\rho$:

$$\begin{cases} \rho > \rho_* & \implies \rho_{new}(\rho) > \rho_* = \rho_{new}(\rho_*), \\ \rho \leq \rho_* & \implies \rho_{new}(\rho) \leq \rho_*, \end{cases} \tag{55}$$

The same inequalities hold true for $\chi = \Xi(\rho, \sigma, \sigma'), \chi_{new} = \Xi(\rho_{new}, \sigma, \sigma')$ and $\chi_* = 1/\varepsilon_*$ as well: if $\chi < \chi_*$, then $\chi_{new} < \chi_*$ and vice versa, since $\Xi(\rho, \sigma, \sigma')$ is monotonously increasing w.r.t. $\rho$.

$\Xi$ **Properties.** In this proof, we omit arguments $\sigma, \sigma'$ of $\Xi^{-1}(\chi, \sigma, \sigma')$ and $\Xi(\rho, \sigma, \sigma')$, because they do not change during IMF step. The second derivative of the function $\Xi(\rho)$ is

$$\frac{d^2\Xi}{d\rho^2}(\rho) = \frac{2\rho(3 + \rho^2)}{\sigma\sigma'(1 - \rho^2)^3}.$$

Hence, we have $\frac{d^2\Xi}{d\rho^2}(\rho) \leq 0$ for $\rho \in (-1, 0]$ and $\frac{d^2\Xi}{d\rho^2}(\rho) \geq 0$ for $\rho \in [0, 1)$. It means that the function $\Xi(\rho)$ is concave on $(-1, 0]$ and convex on $[0, 1)$.

The function $\Xi(\rho)$ is monotonously increasing w.r.t. $\rho$, thus, decreasing of the radius $h \stackrel{\text{def}}{=} |\rho - \rho_*|$ around $\rho_*$ causes the decreasing of $|\chi - \chi_*|$ around $\chi_*$. We consider two cases: $\chi > \chi_*$ and $\chi < \chi_*$.

**Case $\chi > \chi_*$.** We have $\rho = \rho_* + h, \chi = \Xi(\rho_* + h) = \Xi(\rho)$ and $\Xi(\rho_* + \gamma h) \geq \chi_{new}$. We compare the difference using convexity on $[0, 1)$:

$$\begin{aligned} \chi - \chi_{new} &\geq \Xi(\rho_* + h) - \Xi(\rho_* + \gamma h) \geq (\rho_* + h - (\rho_* - h\gamma)) \cdot \frac{d\Xi}{d\rho}(\rho_* + \gamma h) \\ &= (1 - \gamma)h \cdot \frac{d\Xi}{d\rho}(\rho_* + \gamma h). \end{aligned}$$

Since the derivative of $\Xi$ is always positive, we continue the bound:

$$\Xi(\rho_* + h) - \Xi(\rho_* + \gamma h) \geq \min_{\rho' \in [\rho_*, \rho_* + h]} \left| \frac{d\Xi}{d\rho}(\rho') \right| (1 - \gamma)|\rho - \rho_*|.$$

Next, we use Lipschitz property of $\Xi$, i.e.,

$$|\chi - \chi_*| = |\Xi(\rho) - \Xi(\rho_*)| \leq \max_{\rho' \in [\rho_*, \rho_* + h]} \left| \frac{d\Xi}{d\rho}(\rho) \right| |\rho - \rho_*|,$$

and combine it with the previous bound

$$\chi - \chi_{new} \geq \Xi(\rho_* + h) - \Xi(\rho_* + \gamma h) \geq \frac{\min\limits_{\rho' \in [\rho_*, \rho]} \left| \frac{d\Xi}{d\rho}(\rho') \right|}{\max\limits_{\rho' \in [\rho_*, \rho]} \left| \frac{d\Xi}{d\rho}(\rho') \right|} (1 - \gamma)|\chi - \chi_*|.$$

**Case $\chi < \chi_*$.** We have $\rho = \rho_* - h$, $\chi = \Xi(\rho_* - h) = \Xi(\rho)$ and $\Xi(\rho_* - \gamma h) \leq \chi_{new}$. There are three subcases for $\chi, \chi_{new}$ positions around $0$:

1. For positions $\chi_* > \chi_{new} > \chi \geq 0$, we use *convexity* of $\Xi$ on $[0, 1)$ and obtain

$$\chi_{new} - \chi \geq \Xi(\rho_* - \gamma h) - \Xi(\rho_* - h) \geq (1 - \gamma)h \cdot \frac{d\Xi}{d\rho}(\rho_* - h)$$

$$\geq \min_{\rho' \in [\rho_* - h, \rho_*]} \left| \frac{d\Xi}{d\rho}(\rho') \right| (1 - \gamma)|\rho - \rho_*|.$$

2. For positions $\chi_* > 0 \geq \chi_{new} > \chi$, we use *concavity* of $\Xi$ on $(-1, 0]$ and obtain

$$\chi_{new} - \chi \geq \Xi(\rho_* - \gamma h) - \Xi(\rho_* - h) \geq (1 - \gamma)h \cdot \frac{d\Xi}{d\rho}(\rho_* - \gamma h)$$

$$\geq \min_{\rho' \in [\rho_* - h, \rho_*]} \left| \frac{d\Xi}{d\rho}(\rho') \right| (1 - \gamma)|\rho - \rho_*|.$$

3. For positions $\chi_* > \chi_{new} > 0 > \chi$, we use *concavity* of $\Xi$ on $(-1, 0]$ and *convexity* of $\Xi$ on $[0, 1)$ and obtain

$$\chi_{new} - \chi \geq \Xi(\rho_* - \gamma h) - \Xi(\rho_* - h) = [\Xi(\rho_* - \gamma h) - \Xi(0)] + [\Xi(0) - \Xi(\rho_* - h)]$$

$$\geq (\rho_* - \gamma h) \cdot \frac{d\Xi}{d\rho}(0) + (h - \rho_*) \cdot \frac{d\Xi}{d\rho}(0) = (1 - \gamma)h \cdot \frac{d\Xi}{d\rho}(0)$$

$$\geq \min_{\rho' \in [\rho_* - h, \rho_*]} \left| \frac{d\Xi}{d\rho}(\rho') \right| (1 - \gamma)|\rho - \rho_*|.$$

Overall, we make the bound

$$\chi_{new} - \chi \geq \min_{\rho' \in [\rho_* - h, \rho_*]} \left| \frac{d\Xi}{d\rho}(\rho') \right| (1 - \gamma)|\rho - \rho_*|$$

$$\geq \frac{\min\limits_{\rho' \in [\rho, \rho_*]} \left| \frac{d\Xi}{d\rho}(\rho') \right|}{\max\limits_{\rho' \in [\rho, \rho_*]} \left| \frac{d\Xi}{d\rho}(\rho') \right|} (1 - \gamma)|\chi - \chi_*|.$$

For the function $\Xi(\rho) = \frac{\rho}{\sigma_0 \sigma_1 (1 - \rho^2)}$, the centrally symmetrical derivative is

$$\frac{d\Xi}{d\rho}(\rho) = \frac{1 + \rho^2}{\sigma_0 \sigma_1 (1 - \rho^2)^2}.$$

The derivative $\frac{d\Xi}{d\rho}$ has its global minimum at $\rho = 0$. It grows as $\rho \to \pm 1$, hence, the maximum value is achieved at points which are farthest from $0$:

$$\max_{\rho' \in [\rho_*, \rho]} \left| \frac{d\Xi}{d\rho}(\rho') \right| \leq \frac{d\Xi}{d\rho}(\rho),$$

$$\max_{\rho' \in [\rho, \rho_*]} \left| \frac{d\Xi}{d\rho}(\rho') \right| \leq \max \left\{ \frac{d\Xi}{d\rho}(\rho_*), \frac{d\Xi}{d\rho}(|\rho|) \right\},$$

$$\min_{\rho' \in [-1, +1]} \left| \frac{d\Xi}{d\rho}(\rho') \right| \geq \frac{1}{\sigma_0 \sigma_1}.$$

Thus, we prove the bound

$$|\chi - \chi_*| - |\chi_{new} - \chi_*| = |\chi_{new} - \chi| \geq \frac{(1 - \max\{\rho_*, |\rho|\}^2)^2}{1 + \max\{\rho_*, |\rho|\}^2} (1 - \gamma)|\chi - \chi_*|.$$

$$|\chi_{new} - \chi_*| \leq \left[ 1 - (1 - \gamma)\frac{(1 - \max\{\rho_*, |\rho|\}^2)^2}{1 + \max\{\rho_*, |\rho|\}^2} \right] |\chi - \chi_*|.$$

$\square$

### B.4 Proof of IPMF Convergence Conjecture 3.2 for 1-dimensional Gaussians

**Theorem B.5** (Quantitative convergence of IPMF for 1-dimensional Gaussians). *Let $p_0 = \mathcal{N}(\mu_0, \sigma_0^2)$ and $p_1 = \mathcal{N}(\mu_1, \sigma_1^2)$ be 1-dimensional Gaussians. Assume that we run IPMF procedure in the continuous time **or** in discrete time with $N = 1$ intermediate point, starting from some 2-dimensional Gaussian distribution[2]*

$$q^0(x_0, x_1) = \mathcal{N}\left( \begin{pmatrix} \mu_0 \\ \nu \end{pmatrix}, \begin{pmatrix} \sigma_0^2 & \rho_0 \sigma_0 s_0 \\ \rho_0 \sigma_0 s_0 & s_0^2 \end{pmatrix} \right) \text{ with } \rho_0 \in (-1, 1), s_0 > 0$$

*and denote the joint distribution obtained after $k$ IPMF steps by*

$$q^{4k}(x_0, x_1) = \mathcal{N}\left( \begin{pmatrix} \mu_0 \\ \nu_k \end{pmatrix}, \begin{pmatrix} \sigma_0^2 & \rho_k \sigma_0 s_k \\ \rho_k \sigma_0 s_k & s_k^2 \end{pmatrix} \right).$$

*Denote $\chi_k \overset{def}{=} \Xi(\rho_k, \sigma_0, s_k)$. Then the following bounds hold true:*

$$\left| \frac{s_k^2}{\sigma_1^2} - 1 \right| \leq \alpha^{2k} \left| \frac{s_0^2}{\sigma_1^2} - 1 \right|, \qquad |\nu_k - \mu_1| \leq \alpha^k |\nu_0 - \mu_1|, \qquad \left| \chi_k - \frac{1}{\epsilon} \right| \leq \beta^{2k} \left| \chi_0 - \frac{1}{\epsilon_*} \right|.$$

*where factors $\alpha, \beta < 1$ depend on IPMF type (discrete or continuous), initial parameters $s_0, \nu_0, \rho_0$, marginal distributions $p_0, p_1$ and $\epsilon_*$. In particular, $\lim_{k \to \infty} \rho_k = \rho^*$, where $\rho^*$ is the correlation of the static SB solution $q^*$ between $p_0, p_1$, namely, $\rho^* = (\sqrt{\sigma_0^2 \sigma_1^2 + \frac{\epsilon_*^2}{4}} - \frac{\epsilon_*}{2})/(\sigma_0 \sigma_1)$.*

*Proof.* **Notations.** We denote the variance of the $0$-th marginal after the $k$-th IPMF step as $s_k'$. For the first one, we have formula $s_0' = \sqrt{\sigma_0^2 - \sigma_0^2 \tilde{\rho}_0^2 \left( 1 - \frac{\sigma_1^2}{s^2} \right)}$, where $\tilde{\rho}_0$ is the correlation after the first IMF step. More explicitly, $\tilde{\rho}_0 \overset{def}{=} \rho_{new}(\rho_0)$, where $\rho_{new}$ is taken from (46) for continuous IMF and from (48) for DIMF. We denote optimality coefficients $\chi_k \overset{def}{=} \Xi(\rho_k, \sigma_0, s_k)$ and $\chi_* \overset{def}{=} 1/\varepsilon_*$.

**Ranges.** We note that IMF step keeps $s, \nu$, while IPF keeps $\chi$. Due to update equations for $\chi$ (55) and for $s, \nu$ (38), the parameters $s_k, \chi_k$ remain on the same side from $\sigma_1, \frac{1}{\varepsilon_*}$, respectively. Namely, we have ranges for the variances $s_k \in [\sigma_1^{min}, \sigma_1^{max}] \overset{def}{=} [\min\{\sigma_1, s_0\}, \max\{\sigma_1, s_0\}], s_k' \in [\sigma_0^{min}, \sigma_0^{max}] \overset{def}{=} [\min\{\sigma_0, s_0'\}, \max\{\sigma_0, s_0'\}]$ and parameters $\chi_k \in [\chi^{min}, \chi^{max}] \overset{def}{=} [\min\{\chi_*, |\chi_0|\}, \max\{\chi_*, |\chi_0|\}]$.

**Update bounds.** We use update bounds for $\chi$ (54) twice, for $s$ (38) and for $\nu$ (41), however, we need to limit above the coefficients $|\Xi^{-1}(\chi, \sigma, \sigma')|$ and $l(\Xi^{-1}(\chi, \sigma, \sigma'), \Xi^{-1}(\chi_*, \sigma, \sigma'), \gamma(\sigma, \sigma'))$ over the considered ranges of the parameters $\sigma \in [\sigma_0^{min}, \sigma_0^{max}], \sigma' \in [\sigma_1^{min}, \sigma_1^{max}]$ and $\chi \in [\chi^{min}, \chi^{max}]$. The functions $\Xi^{-1}, l, \gamma$ are defined in (43), (54), (44) (or (45) with fixed $t$), respectively.

Since the function $|\Xi^{-1}(\chi, \sigma, \sigma')|$ is increasing w.r.t. $\sigma, \sigma'$ and $\chi$ symmetrically around $0$, we take maximal values $\sigma_0^{max}, \sigma_1^{max}$ and $\chi^{max}$. Similarly, the function

---

[2]We consider $\rho_0 \in (-1, 1)$ only: if $\rho_0 \in \{-1, 1\}$, after the first IMF step, it changes to $\in (-1, 1)$.

$l(\Xi^{-1}(\chi, \sigma, \sigma'), \Xi^{-1}(\chi_*, \sigma, \sigma'), \gamma(\sigma, \sigma'))$ is increasing w.r.t. all arguments symmetrically around 0. Hence, we maximize the function $|\Xi^{-1}|$ and the function $\gamma$, which is also increasing w.r.t. $\sigma$ and $\sigma'$.

**Final bounds.** The final bound after $k$ step of IPMF are:

$$
\begin{aligned}
|s_k^2 - \sigma_1^2| &\leq \alpha^{2k}|s_0^2 - \sigma_1^2|, \\
|\nu_k - \mu_1| &\leq \alpha^k|\nu_0 - \mu_1|, \\
|\chi_k - 1/\varepsilon_*| &\leq \beta^{2k}|\chi_0 - 1/\varepsilon_*|,
\end{aligned}
$$

where $\beta \overset{\text{def}}{=} l(\Xi^{-1}(\chi^{max}, \sigma_0^{max}, \sigma_1^{max}), \Xi^{-1}(\chi_*, \sigma_0^{max}, \sigma_1^{max}), \gamma(\sigma_0^{max}, \sigma_1^{max}))$ and $\alpha \overset{\text{def}}{=} \Xi^{-1}(\chi^{max}, \sigma_0^{max}, \sigma_1^{max})$ taking $l$ from (54), $\gamma$ from (44) for continuous IMF and from (45) with fixed $t$ for discrete IMF. $\square$

## C    ADDITIONAL EXPERIMENTS

### C.1    SB BENCHMARK

We additionally study how well implementations of IPMF procedure starting from different starting processes map initial distribution $p_0$ into $p_1$ by measuring the metric $\mathbb{BW}_2^2$-UVP also proposed by the authors of the benchmark (Gushchin et al., 2023b). We present the results in Table 2. One can observe that DSBM initialized from different starting processes has quite close results and so is the case for ASBM experiments with $\epsilon \in \{1, 10\}$, but with $\epsilon = 0.1$ one can notice that ASBM starting from IPF and *Independent* $p_0 \to p_0$ experience a decline in $\mathbb{BW}_2^2$-UVP metric.

| | | $\epsilon = 0.1$ | | | | $\epsilon = 1$ | | | | $\epsilon = 10$ | | | |
| --- | --- | --- | --- | --- | --- | --- | --- | --- | --- | --- | --- | --- | --- |
| | Algorithm Type | $D=2$ | $D=16$ | $D=64$ | $D=128$ | $D=2$ | $D=16$ | $D=64$ | $D=128$ | $D=2$ | $D=16$ | $D=64$ | $D=128$ |
| Best algorithm on benchmark[†] | Varies | 0.016 | 0.05 | 0.25 | 0.22 | 0.005 | 0.09 | 0.56 | 0.12 | 0.01 | 0.02 | 0.15 | 0.23 |
| DSBM-IMF | | 0.1 | 0.14 | 0.44 | 3.2 | 0.13 | 0.1 | 0.91 | 6.67 | 0.1 | 5.17 | 66.7 | 356 |
| DSBM-IPF | | 0.35 | 0.6 | 0.6 | 1.62 | 0.01 | 0.18 | 0.91 | 6.64 | 0.2 | 3.78 | 81 | 206 |
| DSBM-*Ind* $p_0 \to p_0$ | | 0.08 | 0.38 | 0.62 | 1.72 | 0.13 | 0.18 | 0.84 | 7.45 | 0.04 | 3.72 | 99.3 | 251 |
| SF$^2$M-Sink[†] | IPMF | 0.04 | 0.18 | **0.39** | **1.1** | 0.07 | 0.3 | 4.5 | 17.7 | 0.17 | 4.7 | 316 | 812 |
| ASBM-IMF[†] | | **0.016** | **0.1** | 0.85 | 11.05 | **0.02** | 0.34 | **1.57** | **3.8** | **0.013** | **0.25** | **1.7** | **4.7** |
| ASBM-IPF | | 0.05 | 0.73 | 32.05 | 10.67 | 0.02 | 0.53 | 4.19 | 10.11 | 0.002 | 0.18 | 2.2 | 5.08 |
| ASBM-*Ind* $p_0 \to p_0$ | | 0.36 | 0.76 | 16.33 | 22.63 | 0.07 | 0.48 | 1.93 | 5.36 | 0.04 | 0.23 | 1.04 | 2.29 |

Table 2: Comparisons of $\mathbb{BW}_2^2$-UVP $\downarrow$ (%) between the ground truth static SB solution $p^T(x_0, x_1)$ and the learned solution on the SB benchmark.
The best metric over is **bolded**. Results marked with † are taken from (Gushchin et al., 2024) or (Gushchin et al., 2023b).

### C.2    CELEBA

**SDEdit starting process**.

The IPMF framework doesn't require the starting process to have $p_0, p_1$ marginals or to be a Schrödinger bridge. Then one can try other starting processes that would improve the performance of the IPMF algorithm. Properties of the starting process that would be desirable are 1) $q(x_0) = p_0(x_0)$ and $q(x_1)$ marginal to be close to $p_1(x_1)$ and 2) $q(x_0, x_1)$ to be close to a Schrödinger bridge. In the IMF or IPF we had to choose one of these properties because we can't satisfy both of them completely.

We propose to take a basic image-to-image translation method and use it as a coupling to induce a starting process for the IPMF procedure. Such a coupling would provide 1) $q(x_0) = p_0(x_0)$ and $q(x_1)$ marginal being close to $p_1$ and 2) meaningful pairs between $x_0$ and $x_1$ that would be relatively close to the Schrödinger Bridge. We use SDEdit Meng et al. (2021) which requires an already trained diffusion model (SDE prior) and given an input image $x$, SDEdit first adds noise to the input and then denoises the resulting image by the SDE prior to make it closer to the target distribution of the SDE prior. Various models can be used as an SDE prior, we explore two options: trainable and train-free. As first option we train the DDPM Ho et al. (2020) model on the Celeba 64×64 size female only part and as a second option we take an already trained Stable Diffusion (SD) V1.5 model Rombach et al. (2022) with text prompts conditioned on which model generates 512×512 images similar to the CelebA female part. We then apply SDEdit with the Celeba male images as input to produce

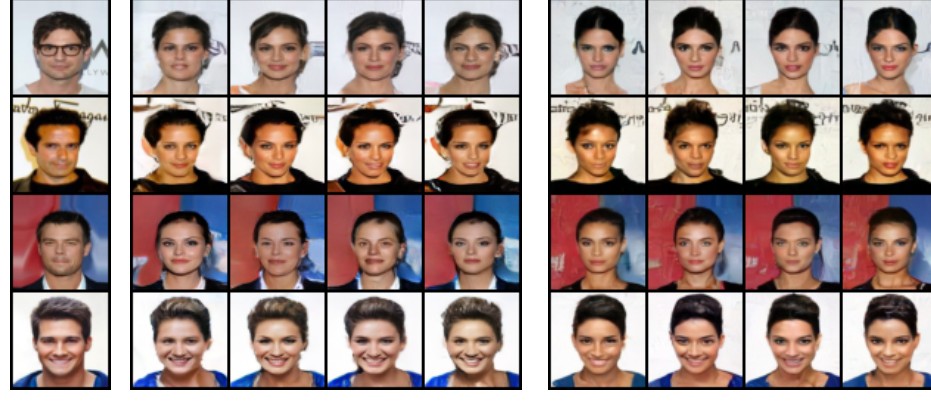

(a) $x \sim p_0$        (b) DSBM-SD SDEdit        (c) DSBM-DDPM SDEdit

Figure 8: Results of CelebA at 64×64 size for *male→female* translation learned with DSBM using SD SDEdit and DDPM SDEdit starting processes for $\epsilon = 1$.

similar female images using trainable DDPM and train-free SDv1.5 approaches, we call the starting processes generated by these SDEdit induced couplings DDPM-SDEdit and SD-SDEdit. SDEdit, DDPM and SDv1.5 hyperparameters are provided in Appendix D.

The visualization of the DSBM implementation of the IPMF procedure starting from *DDPM-SDEdit* and SD-SDEdit processes can be seen in Figure 8 and quantitative evaluation of FID in Figure 6, evaluation of CMMD in Figure 9a and evaluation of MSE in Figure 9b.

**Additional quantitative study.**

In Figure 9, we provide additional quantitative study of IPMF convergence in CMMD (Jayasumana et al., 2024) and Mean Squared Error between inputs and outputs of translation by plotting them as a function of IPMF iteration. Both metrics are calculated on Celeba *male→female* (64 × 64) test set. We notice CMMD plot resembles one of FID, see Figure 6, and MSE plot is approximately the same for the DSBM and ASBM groups.

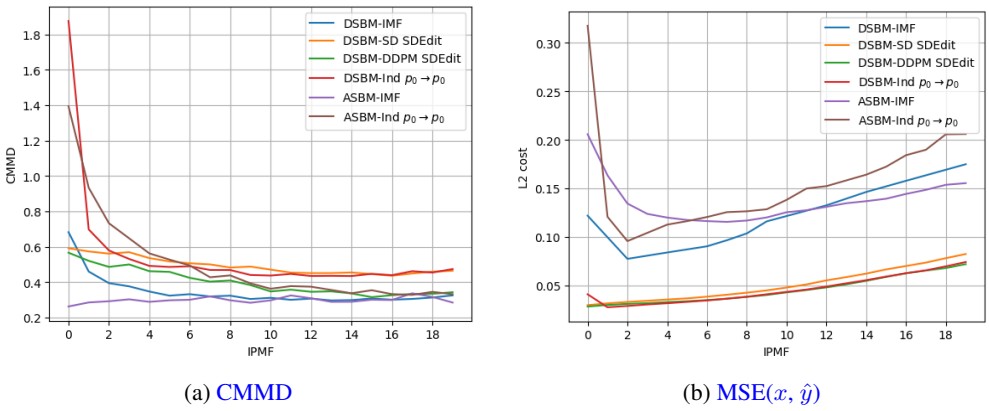

(a) CMMD                    (b) MSE($x, \hat{y}$)

Figure 9: Celeba *male→female* (64 × 64) test set metrics as a function of IPMF iteration for DSBM-IMF, DSBM-*Ind* $p_0 \rightarrow p_0$, DSBM-DDPM SDEdit, DSBM-SD SDEdit, ASBM-IMF, ASBM-*Ind* $p_0 \rightarrow p_0$ is generated by model given $x$ as an input.

## D EXPERIMENTAL DETAILS

### D.1 GENERAL DETAILS

Authors of ASBM (Gushchin et al., 2024) kindly provided us the code for all the experiments. All the hyperparameters including neural networks architectures were chosen as close as possible to the ones used by the authors of ASBM in their experimental section. Particularly, as it is described in

| Model | Dataset | Start process | IPMF iters | IPMF-0 Grad Updates | IPMF-k Grad Updates |
|-------|---------|---------------|------------|---------------------|---------------------|
| ASBM | Celeba | All | 20 | 200000 | 20000 |
| DSBM | Celeba | All | 20 | 100000 | 20000 |
| ASBM | Swiss Roll | All | 20 | 400000 | 40000 |
| DSBM | Swiss Roll | All | 20 | 20000 | 20000 |
| ASBM | cMNIST | All | 20 | 75000 | 38000 |
| DSBM | cMNIST | All | 20 | 100000 | 20000 |
| ASBM | SB Bench | All | 20 | 133000 | 67000 |
| DSBM | SB Bench | All | 20 | 20000 | 20000 |

| Model | Dataset | Start process | NFE | EMA decay | Batch size | D/G opt ratio | Lr G | Lr D |
|-------|---------|---------------|-----|-----------|------------|---------------|------|------|
| ASBM | Celeba | All | 4 | 0.999 | 32 | 1:1 | 1.6e-4 | 1.25e-4 |
| DSBM | Celeba | All | 100 | 0.999 | 64 | N/A | 1e-4 | N/A |
| ASBM | Swiss Roll | All | 4 | 0.999 | 512 | 1:1 | 1e-4 | 1e-4 |
| DSBM | Swiss Roll | All | 100 | N/A | 128 | N/A | 1e-4 | N/A |
| ASBM | cMNIST | All | 4 | 0.999 | 64 | 2:1 | 1.6e-4 | 1.25e-4 |
| DSBM | cMNIST | All | 30 | 0.999 | 128 | N/A | 1e-4 | N/A |
| ASBM | SB Bench | All | 32 | 0.999 | 128 | 3:1 | 1e-4 | 1e-4 |
| DSBM | SB Bench | All | 100 | N/A | 128 | N/A | 1e-4 | N/A |

Table 3: Hyperparameters of models from Celeba 4.4, SwissRoll 4.2, cMNIST 4.4 and SB Bench experiments 4.3. In "Start process" column "All" states for all the used options. "N/A" state for either not used or not applicable corresponding option for the algorithm.

(Gushchin et al., 2024, Appendix D), authors used DD-GAN (Xiao et al.) with Brownian Bridge posterior sampling instead of DDPM's one and implementation from:

```
https://github.com/NVlabs/denoising-diffusion-gan
```

DSBM (Shi et al., 2023) implementation is taken from the official code repository:

```
https://github.com/yuyang-shi/dsbm-pytorch
```

Sampling on the inference stage is done by Euler Maryama SDE numerical solver (Kloeden, 1992) with indicated in Table 3 NFE.

*Independent* $p_0 \to p_0$ starting process in all the experiments was implemented in mini batch manner, i.e., $\{x_{0,n}\}_{n=1}^N \sim p_0$ and $x_1$ batch $\{x_{1,n}\}_{n=1}^N \sim q^0(\cdot|\{x_{0,n}\}_{n=1}^N)$ is generated by permutation of $\{x_{0,n}\}_{n=1}^N$ mini batch indices.

The Exponential Moving Average (EMA) has been used to enhance generator's training stability of both ASBM and DSBM. The parameters of the EMA are provided in Table 3, in case the EMA decay is set to "N/A" no averaging has been applied.

### D.2 ILLUSTRATIVE 2D EXAMPLES

**ASBM**. For toy experiments the MLP with hidden layers $[256, 256, 256]$ has been chosen for both discriminator and generator. The generator takes vector of $(dim+1+2)$ length with data, latent variable and embedding (a simple lookup table `torch.nn.Embedding`) dimensions, respectively. The networks have `torch.nn.LeakyReLU` as activation layer with $0.2$ angle of negative slope. The optimization has been conducted using `torch.optim.Adam` with running averages coefficients $0.5$ and $0.9$. Additionally, the `CosineAnnealingLR` scheduler has been used only at pretraining iteration with minimal learning rate set to 1e-5 and no restarting. To stabilize GAN training R1 regularizer with coefficient $0.01$ (Mescheder et al., 2018) has been used.

**DSBM**. MLP with $[dim + 12, 128, 128, 128, 128, 128, dim]$ number of hidden neurons, `torch.nn.SiLU` activation functions, residual connections between 2nd/4th and 4th/6th layers and Sinusoidal Positional Embedding has been used.

## D.3 SB BENCHMARK

Scrödinger Bridges/Entropic Optimal Transport Benchmark (Gushchin et al., 2023b) and cB$\mathbb{W}_2^2$-UVP, B$\mathbb{W}_2^2$-UVP metric implementation was taken from the official code repository:

https://github.com/ngushchin/EntropicOTBenchmark

Conditional plan metric cB$\mathbb{W}_2^2$-UVP , see Table 1, was calculated over predefined test set and conditional expectation per each test set sample estimated via Monte Carlo integration with 1000 samples. Target distribution fitting metric, B$\mathbb{W}_2^2$-UVP, see Table 2, was estimated using Monte Carlo method and 10000 samples.

**ASBM**. The same architecture and optimizer have been used as in toy experiments D.2, but without the scheduler.

**DSBM**. MLP with $[\dim + 12, 128, 128, 128, 128, 128, \dim]$ number of hidden neurons, `torch.nn.SiLU` activation functions, residual connections between 2nd/4th and 4th/6th layers and Sinusoidal Positional Embedding has been used.

## D.4 CMNIST

Working with MNIST dataset, we use regular train/test split with 60000 images and 10000 images correspondingly. We RGB color train and test digits of classes "2" and "3". Each sample is resized to $32 \times 32$ and normalized by $0.5$ mean and $0.5$ std.

**ASBM**. The cMNIST setup mainly differs by the architecture used. The generator model is built upon the NCSN++ architecture (Song et al.), following the approach in (Xiao et al.) and (Gushchin et al., 2024). We use 2 residual and attention blocks, 128 base channels, and $(1, 2, 2, 2)$ feature multiplications per corresponding resolution level. The dimension of the latent vector has been set to 100. Following the best practices of time-dependent neural networks sinusoidal embeddings are employed to condition on the integer time steps, with a dimensionality equal to $2\times$ the number of initial channel, resulting in a 256-dimensional embedding. The discriminator adopts ResNet-like architecture with 4 resolution levels. The same optimizer with the same parameters as in toy D.2 and SB benchmark D.3 experiments have been used except ones that are presented in Table 3. No scheduler has been applied. Additionally, R1 regularization is applied to the discriminator with a coefficient of 0.02, in line with (Xiao et al.) and (Gushchin et al., 2024).

**DSBM**. The model is based on the U-Net architecture (Ronneberger et al., 2015) with attention blocks, 2 residual blocks per level, 4 attention heads, 128 base channels, $(1, 2, 2, 2)$ feature multiplications per resolution level. Training was held by Adam (Kingma & Ba, 2014) optimizer.

## D.5 CELEBA

Test FID, see Figure 6 is calculated using pytorch-fid package, test CMMD is calculated using unofficial implementation in PyTorch. Working with CelabA dataset (Liu et al., 2015), we use all 84434 male and 118165 female samples ($90\%$ train, $10\%$ test of each class). Each sample is resized to $64 \times 64$ and normalized by $0.5$ mean and $0.5$ std.

**ASBM.** As in cMNIST experiments D.4 the generator model is built upon the NCSN++ architecture (Song et al.) but with small parameter changes. The number of initial channels has been lowered to 64, but the number of resolution levels has been increased with the following changes in feature multiplication, which were set to $(1, 1, 2, 2, 4)$. The discriminator also has been upgraded by growing the number of resolution levels up to 6. No other changes were proposed.

**DSBM**. Following Colored MNIST translation experiment exactly the same neural network and optimizer was used.

**SDEdit coupling**.DDPM Ho et al. (2020) was trained on Celeba female train part processed in the same way as for other Celeba experiments. Number of diffusion steps is equal to 1000 with linear $\beta_t$ noise schedule, number of training steps is equal to 1M, UNet Ronneberger et al. (2015) was used as neural network with 78M parameters, EMA was used during training with rate 0.9999. The DDPM code was taken from the official DDIM Song et al. (2020) github repository:

https://github.com/ermongroup/ddim

The SDEdit method Meng et al. (2021) for DDPM model was used with 400 steps of noising and 400 steps of denoising. The code for SDEdit method was taken from the official github repository:

https://github.com/ermongroup/SDEdit

The Stable Diffusion V1.5 Rombach et al. (2022) model was taken from the Huggingface Wolf et al. (2020) model hub with the tag *"runwayml/stable-diffusion-v1-5"*. The text prompt used is *"A female celebrity from CelebA"*. The SDEdit method implementation for the SDv1.5 model was taken from the Huggingface library Wolf et al. (2020), i.e. *"StableDiffusionImg2ImgPipeline"*, with hyperparameters: *strength* 0.75, *guidance scale* 7.5, *number of inference steps* 50. The output of SDEdit pipeline has been downscaled from 512×512 size to 64×64 size using bicubic interpolation.

## E  BROADER IMPACT.

This paper presents work whose goal is to advance the field of Machine Learning. There are many potential societal consequences of our work, none of which we feel must be specifically highlighted here.

