# OpenReview forum: "Diffusion & Adversarial Schrödinger Bridges via Iterative Proportional Markovian Fitting"
_ICLR.cc/2025/Conference — Submitted to ICLR 2025_

### Official Review · Reviewer_KCc1 · 2024-10-29

**Soundness:** 2
**Presentation:** 3
**Contribution:** 2
**Rating:** 3
**Confidence:** 4

**Summary:**

In this paper, the authors propose a reinterpretation of the Iterative Markovian Fitting (IMF). Iterative Markovian Fitting was proposed in [1, 2]. This scheme performs Markovian and reciprocal projections in order to find the Entropic Optimal Transport (EOT). A much older scheme to find EOT is Iterative Proportional Fitting (IPF) [3, 4]. The authors propose to reinterpret a popular implementation of IMF with forward/backward projections to link this specific IMF implementation with the IPF algorithm. The procedure is called IPMF (Iterative Proportional Markovian Fitting). They show that the proposed procedure converges for one-dimensional Gaussian targets. They then illustrate experimentally the stability of the forward/backward implementation of IMF, i.e. IPMF, for several starting couplings including some that do not satisfy the requirements of IMF and IPF. The experiments are conducted on high dimensional Gaussian, on Colored MNIST and on CelebA.

[1] Shi et al. – Diffusion Schrödinger Bridge Matching

[2] Peluchetti – Diffusion Bridge Mixture Transports, Schrödinger Bridge Problems and Generative Modeling

[3] Kullback – On information and sufficiency, Annals of Mathematics and Statistics

[4] Sinkhorn – A Relationship Between Arbitrary Positive Matrices and Doubly Stochastic Matrices

**Strengths:**

* I think the paper is nicely written. There is proper acknowledgement of prior work and the authors have carefully introduced complex notions such as Iterative Proportional Fitting, Iterative Markovian Fitting, Schrodinger Bridges and Entropic Optimal Transport.

* The authors are also discussing an interesting (and often overlooked) point in the implementation of EOT based samplers which is the initial coupling. While I think the current paper still suffers from major issues, there is value in this study as it puts the emphasis on 1) how stable the IMF is with respect to the original coupling 2) the interaction between IMF and IPF

**Weaknesses:**

* I think the paper has limited novelty. Indeed, the link between the IMF procedure and the IPF was already identified in  [1]. In particular, in [1] the following coupling “We now relate Algorithm 1 to the classical IPF and practical algorithms such as DSB (De Bortoli et al., 2021). Instead of initializing DSBM with $\Pi_{0,T}^0$ given by a coupling between $\pi_0, \pi_1$ , if we initialize it by $\Pi_{0,T}^0 = \mathbb{Q}_{0,T}$ [...] then DSBM also recovers the IPF iterates used in DSB.” Hence the connection between IMF and IPF through the initial coupling was already acknowledged in [1], see Proposition 10 for an explicit statement. In that regard, the IPMF rewriting is not needed to show that the IMF procedure reverts to the IPF one when the coupling is initialized as the IPF one which is one of the main points of the paper.

* Overall I think the paper has limited impact. If it should be read as a theory paper then the result is quite weak (one dimensional Gaussian random variables) and there is not so much insight that is gained from the proof of the theorem, i.e. it seems extremely unclear that the proof generalizes to other settings and what would be a path to reach such general convergence results. If the paper is to be seen as a methodology paper then I think that there should be a clear motivation to choose couplings that are different from the ones of the IMF (i.e. couplings with correct marginals) or the ones from the IPF (couplings with first right marginal and output of the Brownian motion or related stochastic process). So far, I do not see in the paper a clear use case of such couplings. Hence I find the paper poorly motivated.

* Finally, I find the experiments to be quite poor. The only settings that are run are high dimensional Gaussians, 2d examples and unpaired image translation with colored MNIST and Celeba. I found the quantitative study of the last two experiments to be quite poor. For instance, FID scores are presented in Figure 5 and MSE are presented in Appendix C.2 and in that case it seems that changing the coupling yields worse results regarding the quality of the obtained coupling. I think, the authors should investigate other couplings (I know that the minibatch OT one was investigated in the case of Celeba) in order to show that there is something to be gained from the IPMF perspective.

**Questions:**

* Typo in line 59 “Mathcing” should be “Matching”

* Typo in line 374 – “staring” should be “starting”

* In the introduction the authors mention that the reverse Kullback-Leibler between the iterates of the IMF and the target EOT coupling decreases at each step of the IMF. However, I could not find such a result in [1, 2]. To the best of my knowledge, the theoretical study of the IMF is quite limited.

* In Equation (9) and Equation (10) the forward and backward are done successively. Do you think it is possible to learn the backward and forward processes jointly? Is there any work investigating that direction? In that case, it would be interesting to understand the IPMF in that setting.

* It would be good to have some kind of diagram to illustrate the differences between IMF, IPF and IPMF as well as some explicit descriptions on when they are equal. Basically, the choice of the initial coupling matters a lot. If the initial coupling in the IPMF has the right marginals (and is in the reciprocal class) then we recover the IMF. If the initial coupling in the IPMF is Markov in the reciprocal class and has the right initial marginal then we recover the IPF.

* What other starting coupling could be explored? This is related to my comment regarding the lack of applications in the “Weaknesses” section.

* I want to point out that in Section 3.1. The rewriting of the IPMF is only meaningful if we start from a coupling that does not preserve the marginals, otherwise since the Markovian projection preserves the marginal there is no point applying the projection on the last marginal (or the first marginal). I think it is necessary to specify this.

* It would be interesting to investigate what is the effect of approximating the projection in the convergence of the IMF, the IPF and the IPMF, similarly as in [3].

[1] Shi et al. – Diffusion Schrödinger Bridge Matching

[2] Peluchetti – Diffusion Bridge Mixture Transports, Schrödinger Bridge Problems and Generative Modeling

[3] Chen et al. – Provably Convergent Schr ̈odinger Bridge with Applications to Probabilistic
Time Series Imputation

---

> ### Author Response · Authors · 2024-11-23
> **Rebuttal answer. Part I.**
>
> Dear Reviewer KCc1, thank you for your comments. Here are the answers to your questions and comments.
>
> **(1) I think the paper has limited novelty. ... the IPMF rewriting is not needed to show that the IMF procedure reverts to the IPF one when the coupling is initialized as the IPF one which is one of the main points of the paper.**
>
> We agree that the connection between Iterative Markov Fitting (IMF) and Iterative Proportional Fitting (IPF) is a well-established result. However, we would like to clarify that proving this fact is not the primary focus of our work. Instead, we started our discovery by noting that the bidirectional heuristic used in both DSBM and ASBM solvers performs better in practice, while the one-directional training often breaks down due to accumulated errors (Appendix I [1]). We argue that the observed stability arises from the IPF projections "hidden" in this bidirectional procedure. Based on this, we propose a separate IPMF procedure, which generalizes both IPF and IMF procedures. We also make a conjecture that IPMF converges to the Schrödinger Bridge solution for any starting coupling, unlike specific couplings like IPF or IMF. We support our conjecture by providing a rigorous proof of convergence for the one-dimensional Gaussian case (**Section 3.2**) and demonstrating through extensive experiments that convergence is indeed achieved regardless of the choice of initial coupling (**Section 4**).
>
> **(2) If it should be read as a theory paper then the result is quite weak ... If the paper is to be seen as a methodology paper then I think that there should be a clear motivation to choose couplings that are different from the ones of the IMF.**
>
> **Practical part.** From the practical point of view, considering the generalized IPMF procedure opens a door toward using arbitrary starting processes rather than ones that can be used in IPF or IMF procedures. While, in theory, all such starting processes are expected to converge to the same solution, the dynamics of this convergence will differ. Additionally, approximation errors introduced by using neural networks, which differ for different starting processes, will influence the results obtained in practice. We can use this difference introduced by the various couplings to trade-off between focusing on similarity or quality of modeling of target marginal. For instance, in the original version, we considered the independent $p_0 \rightarrow p_0$ starting process, which shows better similarity than other couplings (**Figure 9b**) but worse quality of the target samples (**Figure 6, 9a**).
>
> To further show the practical impact and to answer the related question about what other couplings could be used, we found a new one and conducted experiments with it. Specifically, we found a way to further improve this independent $p_0 \rightarrow p_0$ coupling by improving the quality of target samples. We provide additional experiments on the Celeba dataset with new coupling obtained by using the training-free translation method SDEdit [4] with the pretrained diffusion model (DDPM [5]) for the target distribution. We chose it due to the recent widespread of different pretreated diffusion models used for a wide set of tasks. The coupling provided by SDEdit cannot be used in IMF (since it matches only one of the marginals $p_0$ or $p_1$) and IPF (since it is generated from the Markovian, but not the reciprocal process used in SDEdit). This coupling allows us to obtain a similar quality of generated images as other methods when measured with CMMD (Figure 9a) and slightly worse DSBM-IMF when measured with FID (Figure 6). At the same time, this SDEdit coupling shows a better similarity between the input and generated images than all other couplings except independent $p_0 \rightarrow p_0$. We describe the obtained results and the coupling details in **Section 4.4** and in **Appendix C.2**.
>
> Thus, *we provide an example of how one can design new couplings to trade between the similarity and the quality of target samples or even improve one of the properties without harming the other.* We believe, that it is possible to find even better couplings, for instance, by usage other diffusion models like Stable Diffusion.

---

> ### Author Response · Authors · 2024-11-23
> **Rebuttal answer. Part II.**
>
> **Theoretical part.** From the theoretical point of view, we show that the ideas behind the 1D proof can be generalized to multivariate Gaussians as well. Firstly, we introduce a new *optimality matrix* $A$ that, for a given Gaussian joint distribution, forms a cost function for which this distribution is a EOT solution. During IPF steps, this optimality matrix remains the same, while marginals become exponentially closer to desired values (**Lemmas F.1, F.2**). We believe that IMF step changes the optimality matrix so that it exponentially becomes closer to the matrix of the solution of $\varepsilon$-EOT as we prove in $1D$ case. We verify it in the experiments (**Section 4.1, Figure 2**), where we run IPMF and track the norm between current optimality matrix and $\varepsilon$-EOT matrix.
>
> For more general distributions, we believe that the logic can be saved too. One can use forward KL divergence between the ground truth marginals and current one, since it drops after IPF step and does not change during IMF.  However, one needs only to propose a new optimality parameter instead of optimality matrix which will be kept during IPF step and will be improved after IMF step.
>
> **(3) Finally, I find the experiments to be quite poor. ... For instance, FID scores ... it seems that changing the coupling yields worse results regarding the quality of the obtained coupling. I think, the authors should investigate other couplings ...**
>
> We provided the answer to this in the previous answer.
>
> **(4) In the introduction, the authors mention that the reverse Kullback-Leibler between the iterates of the IMF and the target EOT coupling decreases at each step of the IMF ...**
>
> We meant the reverse Kullback-Leibler between the iterates of the IMF $q^k$ (or $ T^k$ for the continuous-time version) and the ground-truth Schrödinger Bridge process $ q*$ (or $ T*$ for the continuous-time version). Both corresponding proofs of convergence for discrete-time [2, Theorem 3.6] and continuous-time [3, Theorem 8] versions are based on this fact. However, the mentioned fact for the joint distribution on $t=0,t=1$ of IMF iterates ($q^k(x_0, x_1)$) and the target EOT coupling ($q^*(x_0, x_1)$) can be derived from the fact for the processes. We provide the derivation below.
>
> The reverse Kullback-Leibler between the joint distribution on $t=0,t=1$ of IMF iterates ($q^k(x_0, x_1)$) and the target EOT coupling ($q^*(x_0, x_1)$) does not increase at each step of the IMF. Indeed, for any $q^{n}(x_0, x_{\text{in}}, x_1)$ in the discrete IMF sequence, we can write down the following:
> $$
> \text{KL}(q^{n}(x_0, x_{\text{in}}, x_1)|| q^*(x_0, x_{\text{in}}, x_1)) =
> $$
> $$
> \text{KL}(q^{n}(x_0, x_1)|| q^*(x_0, x_1)) + \int \text{KL}({q^{n}(x_{\text{in}}|x_0, x_1)}||q^*(x_{\text{in}}|x_0, x_1))q(x_0,x_1)dx_0 dx_1,
> $$
> $$
> \text{KL}(q^{n}(x_0, x_1)||q^*(x_0, x_1)) =
> $$
> $$
> \text{KL}(q^{n}(x_0, x_{\text{in}}, x_1)|| q^*(x_0, x_{\text{in}}, x_1)) - \int \text{KL}({q^{n}(x_{\text{in}}|x_0, x_1)}||q^*(x_{\text{in}}|x_0, x_1))q(x_0,x_1)dx_0 dx_1,
> $$
> hence
> $$
> \text{KL}(q^{n}(x_0, x_1)||q^*(x_0, x_1)) \leq \text{KL}(q^{n}(x_0, x_{\text{in}}, x_1)|| q^*(x_0, x_{\text{in}}, x_1)).
> $$
> Since $\text{KL}(q^{n}(x_0, x_{\text{in}}, x_1)|| q^*(x_0, x_{\text{in}}, x_1)) \rightarrow 0$ it follows, that $\text{KL}(q^{n}(x_0, x_1)||q^{*}(x_0, x_1)) \rightarrow 0$.
>
> The continuous case can be proved exactly by using the disintegration theorem, as in Eq. 15 in our paper.
>
> **(5)
> In Equation (9) and Equation (10) the forward and backward are done successively. Do you think it is possible to learn the backward and forward processes jointly? ...**
>
> There is an online version of the DSBM algorithm [1], which jointly learns the process given by equations (9) and (10) at the same time. Since this online version shows promising results, it would be interesting to generalize IPMF additionally to support both successive and online (joint training) training. We think it is a good avenue for further research.
>
> **(6)
> It would be good to have some kind of diagram to illustrate the differences between IMF, IPF and IPMF as well as some explicit descriptions on when they are equal ...**
>
> Thank you for your suggestion. To address this, we have added **Figure 1** to the paper, which visually illustrates the differences between Iterative Proportional Fitting (IPF), Iterative Markovian Fitting (IMF), and Iterative Proportional Markovian Fitting (IPMF). The diagram highlights the projection steps for each method and their corresponding properties, such as having the correct marginals, being reciprocal (R), or Markovian (M).
>
> We also added additional clarification at the end of **Section 3.1**, where we defined the IPMF procedure, for which couplings IPMF becomes IPF and for which couplings IPMF becomes IMF.

---

> > ### Author Response · Authors · 2024-11-23
> > **Rebuttal answer. Part III.**
> >
> > **(7)
> > What other starting coupling could be explored? This is related to my comment regarding the lack of applications in the “Weaknesses” section.**
> >
> > We discuss this in detail in the answer to your second question.
> >
> > **(8)
> > I want to point out that in Section 3.1. The rewriting of the IPMF is only meaningful if we start from a coupling that does not preserve the marginals ...**
> >
> > From a theoretical point of view, you are completely right. However, in practice, the Markovian projection could be learned both in forward or backward parametrization, i.e., in the discrete case, we can use the two following variants:
> > $$
> > \text{proj}\_{\mathcal{M}}(q^{2k+1}) \approx \underbrace{q^{2k+1}(x_0)\prod_{n=1}^{N+1}q^{2k+1}\_{\theta}(x_{t_{n}}|x\_{t_{n-1}})}\_{\text{forward parametrization}},
> > $$
> > $$
> > \text{proj}\_{\mathcal{M}}(q^{2k+1}) \approx  \underbrace{q^{2k+1}(x_1)\prod_{n=0}^{N}q^{2k+1}\_{\theta}(x_{t_{n}}|x_{t_{n+1}})}\_{\text{backward parametrization}},
> > $$
> > where $q^{2k+1}\_{\theta}(x_{t_{n}}|x_{t_{n-1}})$ should approximate $q^{2k+1}(x_{t_{n}}|x_{t_{n-1}})$, while $q^{2k+1}\_{\theta}(x_{t_{n}}|x_{t_{n+1}})$ should approximate $q^{2k+1}(x_{t_{n}}|x_{t_{n+1}})$. In the first case, the marginal $q^{2k+1}(x_0)$ will be exactly preserved by construction, while in the second case, the marginal $q^{2k+1}(x_1)$ will be exactly preserved by construction. In turn, due to approximation error, there is no guarantee that the other marginals ($q^{2k+1}(x_1)$ for forward parametrization and $q^{2k+1}(x_0)$ for backward parametrization would be preserved). Thus, even after the first Markovian projections, we most probably end with the coupling, which is Markovian and has only one marginal and not two, as the theory of IMF states. There is even a computational evaluation of this fact for DSBM, showing that usage only of the forward or backward parametrization leads to divergence, while the alternation between then - does not lead to it [1, Appendix I].
> >
> > **(9)
> > It would be interesting to investigate what is the effect of approximating the projection in the convergence of the IMF, the IPF and the IPMF ... .**
> >
> > Thank you for the suggestion. We agree that this is also a promising avenue of research toward better understanding the iterative procedures used to solve the Schrödinger Bridge.
> >
> > **Concluding remarks**.
> > We would be grateful if you could let us know if the explanations we gave have been satisfactory in addressing your concerns and questions about our work. If so, we kindly ask that you consider increasing your rating. We are also open to discussing any other questions you may have.
> >
> > References:
> >
> > [1] De Bortoli V. et al. Schrodinger Bridge Flow for Unpaired Data Translation //The Thirty-eighth Annual Conference on Neural Information Processing Systems.
> >
> > [2] Nikita G. et al. Adversarial Schrödinger Bridge Matching // The Thirty-eighth Annual Conference on Neural Information Processing Systems
> >
> > [3] Shi Y. et al. Diffusion Schrödinger bridge matching //Advances in Neural Information Processing Systems. – 2024. – Т. 36.
> >
> > [4] Meng C. et al. SDEdit: Guided Image Synthesis and Editing with Stochastic Differential Equations //International Conference on Learning Representations.
> >
> > [5] Ho J., Jain A., Abbeel P. Denoising diffusion probabilistic models //Advances in neural information processing systems. – 2020. – Т. 33. – С. 6840-6851.

---

> > > ### Author Response · Authors · 2024-11-27
> > > **Invitation to review our new general response**
> > >
> > > Dear reviewer KCc1, we have uploaded an updated version of the paper to further address your comments regarding the practical impact of the work and usage of different coupling for starting the process of the IPMF procedure. We believe that the new results clearly demonstrate the practical advantages of using the IPMF procedure. We kindly invite you to review our new general response for further details.

---

### Official Review · Reviewer_5NKn · 2024-11-02

**Soundness:** 3
**Presentation:** 3
**Contribution:** 2
**Rating:** 6
**Confidence:** 3

**Summary:**

This paper shows that the popular bidirectional IMF algorithm uses IPF implicitly and they provided a theoretical result proving that for 1d Gaussians, this algorithm IMPF can converges.

**Strengths:**

This paper is well written and shows a novel interesting results on connecting the IMF and IPF with theoretical results for 1d gaussian cases and empirical results for higher dimensional datasets.

**Weaknesses:**

Although this paper shows an interesting theoretical results, it's only for 1d Gaussian cases. I feel like the contribution is relatively weak. If authors can say something more, i.e, for multi-variate Gaussian, this paper will become stronger.

**Questions:**

1. What's the technical difficulties for generalizing the proof to multi-variate Gaussians or other situations?
2. Can you provide more intuition on why IMPF can usually work for any specific starting process?

---

> ### Author Response · Authors · 2024-11-23
> **Rebuttal answer**
>
> Dear Reviewer 5NKn, thank you for your comments. Here are the answers to your questions and comments.
>
> **(1) Although this paper shows an interesting theoretical results, it's only for 1d Gaussian cases. I feel like the contribution is relatively weak. If authors can say something more, i.e, for multi-variate Gaussian, this paper will become stronger.**
>
> We modified the theoretical analysis section and added new conjecture about exponential convergence in higher dimensions, which we back up with new experiments (**Section 4.1**). It completely inherits logic of $1D$ proof. Firstly, we introduce a new *optimality matrix* $A$ that, for a given gaussian joint distribution, forms a cost function for which this distribution is a EOT solution. During IPF steps, this optimality matrix remains the same, while marginals become exponentially closer to desired values (**Lemmas F.1, F.2**). We believe that IMF step changes the optimality matrix so that it exponentially becomes closer to the matrix of the solution of $\varepsilon$-EOT as we prove in $1D$ case. We verify it in the experiments in **Figure 2**, where we run IPMF and track the norm between current optimality matrix and $\varepsilon$-EOT matrix.
>
> **(2) What's the technical difficulties for generalizing the proof to multi-variate Gaussians or other situations?**
>
> We completely generalized analysis of the IPF step for multivariate Gaussians (**Lemmas F.1, F.2**). However, the most difficult part for multivariate Gaussians is the IMF step analysis. We need to show that optimality matrix becomes closer to ground truth solution in some norm. This step is rather technical, i.e., due to very complex matrix transaction equations, it is hard to prove even in a 1D case (but we proved a 1D case in the original version of the paper).
>
> For a more general case, we believe that the logic can be saved. One can use forward KL divergence between the ground truth marginals and current one, since it drops after IPF step and does not change during IMF.  However, one needs to propose  a new optimality parameter instead of optimality matrix which will be kept during IPF step and will be improved after IMF step. Usual for IMF reverse KL divergence between EOT/Schrödinger Bridge solution and current process is not appropriate, since its change during IPF step is unclear.
>
>
> **(3) Can you provide more intuition on why IMPF can usually work for any specific starting process?**
>
> As outlined in **Section 3.1**, the main idea is that the bidirectional IMF procedure combines four types of projections: two IMF-specific projections (reciprocal and Markov) and two IPF-related projections ($p_0$ and $p_1$). Specifically, the IPF projections adjust the marginals $q_0(x_0)$ and $q_1(x_1)$ towards $p_0(x_0)$ and $p_1(x_1)$ but do not influence "optimality" of the process. In turn, the Markov and reciprocal projections adjust the process to make it more optimal (to be Markovian and reciprocal simultaneously) but do not change marginals. Thus, regardless of choosing the initial coupling (it just acts as a starting point) the iterative projections lead to convergence to the Schrödinger Bridge.
>
> **Concluding remarks**.
> We would be grateful if you could let us know if the explanations we gave have been satisfactory in addressing your concerns and questions about our work. If so, we kindly ask that you consider increasing your rating. We are also open to discussing any other questions you may have.

---

> > ### Author Response · Authors · 2024-11-29
> >
> > Dear reviewer 5NKn,
> >
> > We thank you for your review and appreciate your time reviewing our paper.
> >
> > The end of the discussion period is close. We would be grateful if we could hear your feedback regarding our answers to the reviews. We are happy to address any remaining points during the remaining discussion period.
> >
> > Thanks in advance,
> >
> > Paper authors

---

> > > ### Author Response · Authors · 2024-12-02
> > >
> > > Dear reviewer 5NKn,
> > >
> > > We thank you for your review and appreciate your time reviewing our paper.
> > >
> > > The end of the discussion period is close. We would be grateful if we could hear your feedback regarding our answers to the reviews. We are happy to address any remaining points during the remaining discussion period.
> > >
> > > Thanks in advance,
> > >
> > > Paper authors

---

### Official Review · Reviewer_tu1b · 2024-11-02

**Soundness:** 3
**Presentation:** 3
**Contribution:** 3
**Rating:** 6
**Confidence:** 3

**Summary:**

The paper targets the solution of the Schrodinger Bridge problem. It first revisits two standard methods to solve the problem: Iterative Markovian Fitting (IMF) and Iterative Proportional Fitting (IPF). Then, the authors identify a connection between both methods and the associated heuristics of their implementation. This leads them to propose the Iterative Proportional Markovian Fitting (IPMF). The authors present a theoretical formulation and analysis of IPMF, to then provide experimental validation over synthetic and real-world image datasets.

**Strengths:**

The paper is clear, and in the first pages it adopts a tutorial-like style. The presentation of the relevant background (IPF & IMF) is properly achieved and the proposal is delivered in context. The authors also prove a particular case of their proposal (Thm 3.2).

Experimentally, the paper includes empirical validation on synthetic (Gaussians of different dimensions) and image datasets (Celeba, MNIST). The experiments are complemented by both quantitative and qualitative performance evaluations.

**Weaknesses:**

There are some inaccuracies in the text (Appenix, markovian, Matkovian, etc).

Perhaps due to the lack of familiarity of this reviewer with the SOTA on SB: the main contribution of the paper is the identification that IPF & IMF can be understood as the same procedure. However, what are the practical implications of this? Is this a a purely conceptual statement, or does this fact help practical applications of SB? Perhaps the authors can be more specific about the practical advantages of their proposal

**Questions:**

please see above

---

> ### Author Response · Authors · 2024-11-23
> **Rebuttal answer**
>
> Dear Reviewer tu1b, thank you for your comments. Here are the answers to your questions and comments.
>
> **(1) There are some inaccuracies in the text (Appenix, markovian, Matkovian, etc).**
>
> Thank you for pointing out this. We fixed these and several similar typos in the new revision.
>
> **(2) Perhaps due to the lack of familiarity of this reviewer with the SOTA on SB: the main contribution of the paper is the identification that IPF & IMF can be understood as the same procedure. However, what are the practical implications of this? Is this a  purely conceptual statement, or does this fact help practical applications of SB? Perhaps the authors can be more specific about the practical advantages of their proposal.**
>
> From the practical point of view, considering the generalized IPMF procedure opens a door toward using arbitrary starting processes rather than ones that can be used in IPF or IMF procedures. While, in theory, all such starting processes are expected to converge to the same solution, the dynamics of this convergence will differ. Additionally, approximation errors introduced by using neural networks, which differ for different starting processes, will influence the results obtained in practice. We can use this difference introduced by the various couplings to trade-off between focusing on similarity or quality of modeling of target marginal. For instance, in the original version, we considered the independent $p_0 \rightarrow p_0$ starting process, which shows better similarity than other couplings (**Figure 9b**) but worse quality of the target samples (**Figure 6, 9a**).
>
> To further show the practical impact, we found a new one and conducted experiments with it. Specifically, we found a way to further improve this independent $p_0 \rightarrow p_0$ coupling by improving the quality of target samples. We provide additional experiments on the Celeba dataset with new coupling obtained by using the training-free translation method SDEdit [1] with the pretrained diffusion model (DDPM [2]) for the target distribution. We chose it due to the recent widespread of different pretreated diffusion models used for a wide set of tasks. The coupling provided by SDEdit cannot be used in IMF (since it matches only one of the marginals $p_0$ or $p_1$) and IPF (since it is generated from the Markovian, but not the reciprocal process used in SDEdit). This coupling allows us to obtain a similar quality of generated images as other methods when measured with CMMD (**Figure 9a**) and slightly worse DSBM-IMF when measured with FID (**Figure 6**). At the same time, this SDEdit coupling shows a better similarity between the input and generated images than all other couplings except independent $p_0 \rightarrow p_0$. We describe the obtained results and the coupling details in **Section 4.4** and in **Appendix C.2**.
>
> Thus, *we provide an example of how one can design new couplings to trade between the similarity and the quality of target samples or even improve one of the properties without harming the other.* We believe, that it is possible to find even better couplings.
>
> **Concluding remarks**.
> We would be grateful if you could let us know if the explanations we gave have been satisfactory in addressing your concerns and questions about our work. If so, we kindly ask that you consider increasing your rating. We are also open to discussing any other questions you may have.
>
> References:
>
> [1] Meng C. et al. SDEdit: Guided Image Synthesis and Editing with Stochastic Differential Equations //International Conference on Learning Representations.
>
> [2] Ho J., Jain A., Abbeel P. Denoising diffusion probabilistic models //Advances in neural information processing systems. – 2020. – Т. 33. – С. 6840-6851.

---

> > ### Author Response · Authors · 2024-11-27
> > **Invitation to review our new general response**
> >
> > Dear reviewer tu1b, we have uploaded an updated version of the paper to further address your comments regarding the practical impact of the work. We believe that the new results clearly demonstrate the practical advantages of using the IPMF procedure. We kindly invite you to review our new general response for further details.

---

### Official Review · Reviewer_BjZj · 2024-11-04

**Soundness:** 3
**Presentation:** 3
**Contribution:** 2
**Rating:** 6
**Confidence:** 2

**Summary:**

In the context of Schrödinger Bridge (SB) problems, the paper analyses theoretically some heuristic algorithms typically used in practice in the literature. Specifically, the authors give a new interpretation to these heuristics, as a combination of well-known procedures used in the context of theoretical analysis of SB problems. The authors name the resulting procedure "Iterative Proportional Markov Fitting" (IPMF).

The main result of the paper is to show the converge of IPMF when both the distributions $p_0$ and $p_1$ are unidimensional gaussians.
Furthermore, the authors study heuristically the convergence of IPMF in more interesting settings, by performing numerical experiments on some typical datasets and benchmarks.

**Strengths:**

- The paper is very well written.
- The problem in consideration and the background are introduced very clearly, so that the idea of the paper is easy to comprehend.
- The proof for 1D gaussian, although limited, could lead to generalisations that improve on the results that are available in the literature at the moment.

**Weaknesses:**

- Section 4, presenting the experiments, seems to be a bit disconnected from the previous sections. While at the beginning the authors say the aim of the section is to show the convergence of the different procedures in more generic settings, the focus shifts on the comparison between the performance of different procedures. Keeping the focus on the convergence of the procedures to the target could enhance the readability of the section.

[Minor]
Typos:
- Line 172: Matkovian
- Line 374: staring
- Line 412: Additionaly
- Line 417: Scrödinger
- Line 528: Appenix

A careful reading of the paper is suggested to remove additional typos.

**Questions:**

- In section 4.3, the results presented in Table 1 are difficult to interpret. What's the role of the "volatility" parameter $\epsilon$ and why it influences so much the performances of the different procedures?

---

> ### Author Response · Authors · 2024-11-23
> **Rebuttal answer. Part I.**
>
> Dear Reviewer BjZj, thank you for your comments. Here are the answers to your questions and comments.
>
> **(1) Section 4, presenting the experiments, seems to be a bit disconnected from the previous sections. ...**
>
> Thank you for this comment. This impression likely arises because, in the first experimental setup (“High Dimensional Gaussians”), the discussion emphasizes differences in convergence speed. We agree that we should better highlight that the main observation is the algorithm’s convergence to the ground-truth solution across all couplings. **We have adjusted the text in this section** (**Section 4**) to stress this point more clearly.
>
> We also would like to note that we follow the initial aim stated at the beginning of **Section 4** and consistently conclude the discussions in other experiments with observations of convergence, as seen in the following lines:
>
> - “In all the cases, we observe convergence in the target distribution.” (line 418)
>
> - “… yield quite similar results.” (line 430)
>
> - “We observe that both DSBM and ASBM algorithms starting from both IMF and Inverted 7 starting process fit the target distribution of colored MNIST digits of class ‘2’ and preserve the color of the input image during translation.” (lines 458–460)
>
> - “We see from the qualitative results in **Figure~6** that all trained models: 1) converge to the target distribution, 2) keep alignment between the features of the input images and generated images (e.g., the color of hair, background, etc.).” (lines 484–485)
>
> If you are referring to the ASBM and DSBM algorithms (i.e., discrete-time and continuous-time approaches to solving Schrödinger Bridge) by “different procedures,” we would like to clarify that we included both to demonstrate that the IPMF generalization is applicable to both frameworks.
>
> To resolve this weakness further, **we additionally added** in the beginning of the experimental section (**Section 4**) that our goal is to achieve the same or similar results for all used starting coupling and for both discrete-time (ASBM) and continuous-time (DSBM) solvers.
>
> **(2) [Minor] Typos**
>
> Thank you for pointing out this. We fixed these and several similar typos in the new revision.

---

> > ### Author Response · Authors · 2024-11-23
> > **Rebuttal answer. Part II.**
> >
> > **(3) In section 4.3, the results presented in Table 1 are difficult to interpret. What's the role of the "volatility" parameter and why it influences so much the performances of the different procedures?**
> >
> > From the point of view of the dynamic Schrödinger Bridge (i.e., not static), the parameter $\sqrt{\epsilon}$ is the variance of the reference Wiener prior process $W^{\epsilon}$, defined via the stochastic differential equation:
> > $$
> > dx_t = \sqrt{\epsilon} dW_t.
> > $$
> > This parameter regulates the stochasticity of the trajectories of the solution of the Schrödinger Bridge $q^*(x_0, x_{\text{in}}, x_1)$ in the discrete case or $T^*$ in the continuous case. For instance, the solution in the continuous case is given by the SDE:
> > $$
> > dx_t = v^*(x_t, t)dt + \sqrt{\epsilon} dW_t, \quad x_0 \sim p_0(x_0),
> > $$
> > i.e., the noise "injected" during the trajectory is proportional to $\sqrt{\epsilon}$.
> >
> > From the point of view of the static Schrödinger Bridge problem, which has the functional:
> > $$
> > \text{KL}(q(x_0, x_1) || p^{W^{\epsilon}}(x_0, x_1)) =  \int \frac{||x_1 - x_0||^{2}}{2\epsilon}dq(x_0, x_1) - H(q(x_0, x_1)) + C,
> > $$
> > the coefficient $\epsilon$ is a weight that balances between the quadratic cost $\frac{\|x_1 - x_0\|^2}{2\epsilon}$ and the entropy of the solution $H(q(x_0, x_1))$. Larger values of $\epsilon$ lead to a solution with higher entropy (i.e., greater variability of samples and less similarity), and vice versa.
> >
> > In **Table 1**, we follow the authors of the benchmark and provide the metric of how well different algorithms restored the ground-truth solution $q^*(x_0, x_1)$ of the static Schrödinger Bridge. The values $\epsilon = 0.1, 1.0, 10$ chosen by the authors of the benchmark roughly correspond to "low," "moderate," and "high" values.
> >
> > Typically, continuous-time methods (like DSBM [1], SF$^2$M-Sink [2]) have difficulties with high values of $\epsilon = 10$, since to precisely simulate the SDE:
> > $$
> > dx_t = v^*(x_t, t)dt + \sqrt{\epsilon} dW_t, \quad x_0 \sim p_0(x_0),
> > $$
> > one needs to use many steps of numerical integration and perfectly learn the drift $v^*$. Otherwise, due to the high variance of the noise injected during solving, this SDE approximation quickly accumulates errors and leads to divergence. However, the precise fitting of $v^*$ is hard since the variance of the objective of such methods usually grows with the values of $\epsilon$.
> >
> > In turn, the performance of discrete-time methods such as ASBM does not show such dependence on the coefficient $\epsilon$ since they learn only the transitional density and not the drift function. However, for low values of $\epsilon$, due to adversarial training, their performance can be worse than their continuous-time counterparts.
> >
> > **Concluding remarks**.
> > We would be grateful if you could let us know if the explanations we gave have been satisfactory in addressing your concerns and questions about our work. If so, we kindly ask that you consider increasing your rating. We are also open to discussing any other questions you may have.
> >
> > References:
> >
> > [1] Shi Y. et al. Diffusion Schrödinger bridge matching //Advances in Neural Information Processing Systems. – 2024. – Т. 36.
> >
> > [2] Tong A. et al. Simulation-Free Schrödinger Bridges via Score and Flow Matching //ICML Workshop on New Frontiers in Learning, Control, and Dynamical Systems.

---

> > > ### Comment · Reviewer_BjZj · 2024-11-26
> > >
> > > I thank the authors for addressing my doubts and questions.
> > > Having also read the other reviews and the authors' responses to them, I decided to maintain my positive rating.

---

### Author Response · Authors · 2024-11-23
**General response**

Dear reviewers, thank you all for reviewing our paper and for your insightful comments. We appreciate that all of you found our work well-written (BjZj, tu1b, 5NKn, KCc1) and that you recognized our theoretical contributions toward better understanding and generalizing existing SB procedures like IMF and IPF (BjZj, KCc1, 5NKn). Additionally, we are pleased that you highlighted our empirical evaluations of the IPMF procedure under diverse settings as the strength (5NKn, tu1b). Please find the answers to your questions below.

**Revised paper.** We uploaded a revised version of our paper. To sum up, the main edits to the paper are highlighted with the **blue color** (newly added) and **orange color** (additional highlighting of the most important parts).

- [**reviewers: KCc1, 5NKn**] We modified the theoretical part of our work (**Section 3.2**, **Appendix B**). Specifically, we added a new conjecture about exponential convergence in higher dimensions with a detailed motivation. We generalized the analysis of the IPF step for multivariate Gaussians, while the IMF step analysis remains rather technical than conceptual challenge.

- [**reviewers: KCc1, 5NKn**] We modified the setup for the high-dimensional experiments by increasing the number of intermediate points $N$ from $1$ to $3$ and dimensionality from $D=16$ to $D=128$ to check the more general case of our conjecture. We also provided more plots (**Section 4.1**) to support our modified and extended theoretical part on the convergence rate of IPMF for multivariate Gaussian

- [**reviewers: KCc1, tu1b**] We added the experiment with a new coupling on the CelebA dataset (**Section 4.4**, **Appendix C.2**), obtained by using training-free translation method SDEedit [1] with pretrained diffusion model [2].

- [**reviewers: KCc1, tu1b**] To ensure better comparison, we reran and updated all the results for DSBM with independent $p_0 \rightarrow p_0$ starting process using the $100000$ gradient updates (as we use for DSBM-IMF) instead of $20000$ as we used in the first version (**Section 4.4**). The new results do not significantly differ from the previous ones.

- [**reviewers: KCc1, tu1b**] We recomputed the MSE  for the DSBM solvers due to the initial computation error. Now, the ASBM and DSBM solvers exhibit similar MSE metrics and differ mainly by the used coupling (**Figure 7b**).

- [**reviewer: KCc1**] We introduce a new diagram (**Figure 1**) to visually clarify the relationships and distinctions between IPF, IMF, and discovered IPMF procedures..

References:

[1] Meng C. et al. SDEdit: Guided Image Synthesis and Editing with Stochastic Differential Equations //International Conference on Learning Representations.

[2] Ho J., Jain A., Abbeel P. Denoising diffusion probabilistic models //Advances in neural information processing systems. – 2020. – Т. 33. – С. 6840-6851.

---

> ### Author Response · Authors · 2024-11-25
>
> Dear reviewers,
>
> We thank you for your review and appreciate your time reviewing our paper.
>
> The end of the discussion period is close. We would be grateful if we could hear your feedback regarding our answers to the reviews. We are happy to address any remaining points during the remaining discussion period.
>
> Thanks in advance,
>
> Paper authors

---

> > ### Author Response · Authors · 2024-11-27
> > **General response II**
> >
> > Dear reviewers, we have uploaded an updated version of the paper. The updates further address the points regarding the practical impact of our paper and explore other possible couplings for starting processes (reviewers: **KCc1, tu1b**). The newly added content, as previously, is highlighted in blue. The changes include:
> >
> > - [**reviewers: KCc1, tu1b**] We added the new experiment with a new coupling on the CelebA dataset (**Section 4.4**, **Appendix C.2**), obtained by using training-free translation method SDEedit [1] with **Stable Diffusion** model [3].
> >
> > The new DSBM model trained with the IPMF coupling based on SDEdit and Stable Diffusion (DSBM-SD SDEdit) provides **significantly better similarity** (**Figure 6, Figure 8**) between the input and generated images compared to the DSBM with the classical IMF coupling. Moreover, the **sample quality** of the new DSBM model (DSMB-SD SDEdit) **is comparable** (according to the standard FID metric) with the DSBM model trained with the classical IMF coupling (**Figure 6, Figure 8**), which has the best sample quality among DSBM models.
> >
> > Notably, as with the coupling produced by SDEdit using DDPM [2] pre-trained for the target distribution, this new coupling cannot be used in IMF (since it matches only one of the marginals) and IPF (since it is generated from the Markovian, but not the reciprocal process used in SDEdit).
> >
> > Unlike the previously reported SDEdit with DDPM coupling, the Stable Diffusion is given off the shelf in this case, i.e., one does not need to train a new diffusion model for the specific target dataset each time, just to download an open-source neural network and run SDEedit with it.
> >
> > Thus, we introduced a way to improve the results of classical DSBM used with IMF coupling, which is known as one of the best approaches to the SB problem. *We believe these results support our position that the IPMF procedure has clear practical impact.*
> >
> > References:
> >
> > [1] Meng C. et al. Sdedit: Guided image synthesis and editing with stochastic differential equations //arXiv preprint arXiv:2108.01073. – 2021.
> >
> > [2] Ho J., Jain A., Abbeel P. Denoising diffusion probabilistic models //Advances in neural information processing systems. – 2020. – Т. 33. – С. 6840-6851.
> >
> > [3] Esser P. et al. Scaling rectified flow transformers for high-resolution image synthesis //Forty-first International Conference on Machine Learning. – 2024.

---

### Meta-Review · Area_Chair_GVgy · 2024-12-21

**Metareview:**

(a) Summary of the paper's claims and findings:

The paper introduces the Iterative Proportional Markovian Fitting (IPMF) procedure, combining aspects of the Iterative Markovian Fitting (IMF) and Iterative Proportional Fitting (IPF) algorithms. The authors theoretically demonstrate convergence of the IPMF procedure in the limited case of one-dimensional Gaussian distributions and present empirical results for high-dimensional Gaussian distributions, MNIST, and CelebA datasets. They argue that IPMF generalizes prior methods and offers practical flexibility for starting couplings, showing stability in broader settings.

(b) Strengths:

The paper is well-written, providing clear explanations of foundational methods (IMF, IPF) and their connections.
It addresses an important aspect of stability in Schrödinger Bridge implementations.
Experimental results are provided to support practical utility, including a new coupling derived using SDEdit.
The theoretical insight into the convergence for 1D Gaussian cases is rigorous and includes experimental support for broader scenarios.
(c) Weaknesses:

The theoretical results are limited to 1D Gaussian cases, offering little insight into generalization to higher dimensions or other distributions.
The novelty is weak, as prior work has already connected IMF and IPF. The proposed IPMF framework does not significantly extend the theoretical or practical frontier.
Experimental evaluations are narrow and lack comprehensive comparisons with state-of-the-art methods. While some new couplings are explored, the practical gains are marginal or unclear.
The methodology lacks a compelling motivation or application scenario where IPMF is preferable over existing techniques.

(d) Reasons for rejection:

The paper’s contribution is found to be incremental, with limited theoretical novelty and insufficient practical impact. The proposed framework reiterates known connections and does not provide a strong rationale for adopting IPMF in place of established methods. Despite the authors' responses and updates, the scope of the results and their applicability remain constrained.

**Additional Comments On Reviewer Discussion:**

During the rebuttal period, the authors provided extensive responses, including a revised manuscript addressing several reviewer concerns:

Reviewer KCc1 questioned the novelty of the IPMF framework and the practicality of using alternative couplings. The authors clarified motivations and added experiments.
Reviewer 5NKn suggested expanding theoretical results to multivariate Gaussians. While the authors introduced a conjecture and experimental evidence, the absence of a generalized proof limits the contribution's depth.
Reviewer tu1b and BjZj noted that the experimental sections lacked focus and specificity. Updates provided additional explanations but failed to substantively strengthen the empirical analysis.

While the authors made commendable efforts to improve clarity and address weaknesses, the core contributions remain insufficient for acceptance. KCc1's primary concerns—limited novelty, narrow theoretical scope, and modest practical impact—persist in the revised submission.

---

### Decision · Program_Chairs · 2025-01-22

Reject